# Staying up to Date with Online Content Changes Using Reinforcement Learning for Scheduling

**Andrey Kolobov** [1]   **Yuval Peres** [2]   **Cheng Lu** [3]   **Eric Horvitz** [4]

## Abstract

From traditional Web search engines to virtual assistants and Web accelerators, services that rely on online information need to continually keep track of remote content changes by explicitly requesting content updates from remote sources (e.g., web pages). We propose a novel optimization objective for this setting that has several practically desirable properties, and efficient algorithms for it with optimality guarantees even in the face of mixed content change observability and initially unknown change model parameters. Experiments on 18.5M URLs crawled daily for 14 weeks show significant advantages of this approach over prior art.

## 1. Introduction

As the Web becomes more and more dynamic, services that rely on web data face the increasingly challenging problem of keeping up with online content changes. Whether it be a continuous-query system (Pandey et al., 2003), a virtual assistant like Cortana or Google Now, or an Internet search engine, such a service tracks many remote sources of information – web pages or data streams (Pandey et al., 2004). Users expect these services, which we call *trackers*, to be surface the latest information that appears at the sources. This is easy to do when the remote sources *push* content updates to the tracker. Unfortunately, the push model is impractical for maintaining databases as large as major search engines' indexes (Baeza-Yates & Ribeiro-Neto, 1999) and conversation agents' knowledge graphs (Hixon et al., 2015). Instead, for all sources they track, these services must continually decide when to re-request (*crawl*) their data in order pick up the changes. A policy that makes these decisions

[1]akolobov@microsoft.com, Microsoft Research, Redmond, WA, USA [2]At Microsoft Research at the time of work on this publication [3]Cheng.Lu@microsoft.com, Microsoft Bing, Bellevue, WA, USA [4]horvitz@microsoft.com, Microsoft Research, Redmond, WA, USA. Correspondence to: Andrey Kolobov <akolobov@microsoft.com>.

*Reinforcement Learning for Real Life (RL4RealLife) Workshop in the 36th International Conference on Machine Learning*, Long Beach, California, USA, 2019. Copyright 2019 by the author(s).

well solves *freshness crawl scheduling problem*.

Freshness crawl scheduling has several challenging aspects. For most sources, the tracker finds out whether they have changed only when it crawls them. To guess when the changes happen, and hence should be downloaded, the tracker needs a predictive model whose parameters are initially unknown. Thus, the tracker needs to learn these models *and* optimize a freshness-related objective when scheduling crawls. For some web pages, however, sitemap polling and other means can provide trustworthy near-instantaneous signals that the page has changed in *a* meaningful way, though not what the change is exactly. But even with these remote change observations and known change model parameters, freshness crawl scheduling remains highly nontrivial because the tracker cannot react to every individual predicted or actual change. The tracker's infrastructure imposes a *bandwidth constraint* on the average daily number of crawls, usually limiting it to a fraction of the change event volume. Last but not least, Google and Bing track many billions of pages (van den Bosch et al., 2015) with vastly different importance and change frequency characteristics. The sheer size of this constrained learning and optimization problem makes low-polynomial algorithms strongly preferable, despite the availability of big-data platforms.

This paper presents a holistic approach to freshness crawl scheduling that handles all of the above aspects in a computationally efficient manner with optimality guarantees using a type of reinforcement learning (Sutton & Barto, 1998). Different aspects of crawl scheduling have been studied extensively before, as described in detail in the Related Work section. Crawl schedule optimization alone, under various objectives and assuming known model parameters, has been the focus of many papers since Coffman et al. (1998). However, even without remote change sensing and under the known-model assumption, efficient optimization has so far been possible only for freshness objectives that induce policies with practically undesirable behaviors such as crawl-starving many of the sources forever (Azar et al., 2018). Learning change-predictive models has received much attention as well (e.g., (Cho & Garcia-Molina, 2003b)) but purely as a preprocessing step, without regard to the learning versus schedule optimization tradeoff. No prior work has studied joint optimization of all facets of freshness crawl scheduling that we consider here. Specifically, our contributions are as follows:

- We propose a natural freshness optimization objective based on harmonic numbers, and show how its mathematical properties enable efficient optimal scheduling.

- We derive efficient optimization procedures for this bandwidth-constrained objective under complete, mixed, and lacking remote change observability.

- We present a reinforcement learning algorithm that integrates these approaches with model estimation of Cho & Garcia-Molina (2003b) and converges to the optimal policy, lifting the known-parameter assumption.

- We introduce an approximate crawl scheduling algorithm that requires learning far fewer parameters, and derive a condition under which its solution is optimal.

## 2. Problem formalization

In settings we consider, a service we call *tracker* monitors a set $W$ of information sources. A source $w \in W$ can be a web page, a data stream, a file, etc, whose content occasionally changes. To pick up changes from a source, the tracker needs to *crawl* it, i.e., download its content. When source $w$ has changes the tracker hasn't picked up, the tracker is *stale* w.r.t. $w$; otherwise, it is *fresh* w.r.t. $w$. We assume near-instantaneous crawl operations, and a fixed set of sources $W$. Growing $W$ to improve *information completeness* (Pandey et al., 2004) is also an important but distinct problem; we do not consider it here.

**Discrete page changes.** We define a *content change* at a source as an alteration at least minimally important to the tracker. In practice, trackers compute a source's content *digest* using data extractors, *shingles* (Broder et al., 1997), or *similarity hashes* (Charikar, 2002), and consider content changed when its digest changes.

**Models of change process and importance.** We model each source $w \in W$'s changes as a Poisson process with *change rate* $\Delta_w$. Many prior works adopted it for web pages (Cho & Garcia-Molina, 2000b; Cho & Ntoulas, 2002; Cho & Garcia-Molina, 2003a;b; 2000a; Wolf et al., 2002; Azar et al., 2018) as a good balance between fidelity and computational convenience. We also associate an *importance* score $\mu_w$ with each source, and denote these parameters jointly as $\vec{\mu}$. Importance score $\mu_w$ can be thought of as characterizing the time-homogeneous Poisson rate at which the page is served in response to the query stream, although in general it can be any positive weight measuring source significance (Azar et al., 2018). ***While scores $\mu_w$ are defined by, and known to, the tracker, change rates $\Delta_w$ need to be learned.***

**Change observability.** For most sources, the tracker can find out whether the source has changed only by crawling it. In this case, even crawling doesn't tell the tracker how many times the source has changed since the last crawl. We denote the set of these sources as $W^-$ and say that the

tracker receives *incomplete change observations* about them. However, for other sources, which we denote as $W^o$, the tracker may receive near-instant notification whenever they change, i.e., get *complete remote change observations*. E.g., for web pages these signals may be available from browser telemetry or sitemaps. Thus the tracker's set of sources can be represented as $W = W^o \cup W^-$ and $W^o \cap W^- = 0$.

**Bandwidth constraints.** Even if the tracker receives complete change observations, it generally cannot afford to do a crawl upon each of them. The tracker's network infrastructure and considerations of respect to other Internet users limit its *crawl rate* (the average number of requests per day); the total *change rate* of tracked sources may be much higher. We call this limit *bandwidth constraint $R$*.

**Optimizing freshness.** The tracker operates in continuous time and starts fresh w.r.t. all sources. Our scheduling problem's solution is a policy $\pi$ — a rule that at every instant $t$ chooses (potentially stochastically) a source to crawl or decides that none should be crawled. Executing $\pi$ produces a *crawl sequence* of time-source pairs $CrSeq = (t_1, w_1), (t_2, w_2), \ldots$, denoted $CrSeq_w = (t_1, w), (t_2, w), \ldots$ for a specific source $w$. Similarly, the (Poisson) change process at the sources generates a change sequence $ChSeq = (t_1', w_1'), (t_2', w_2'), \ldots$, where $t_i'$ is a change time of source $w_i'$; its restriction to source $w$ is $ChSeq_w$. We denote the joint process governing changes at all sources as $P(\vec{\Delta})$.

## 3. Minimizing harmonic staleness penalty

We view maximizing freshness as minimizing costs the tracker incurs for the lack thereof, and associate the following *policy cost $J^\pi$* with every scheduling policy $\pi$, the *time-averaged expected staleness penalty*:

$$J^\pi = \lim_{T \to \infty} \mathop{\mathbb{E}}_{\substack{CrSeq \sim \pi, \\ ChSeq \sim P(\vec{\Delta})}} \left[ \frac{1}{T} \int_0^T \sum_{w \in W} \mu_w C(N_w(t)) dt \right]$$
(1)

Here, $T$ is a planning horizon, $N_w(t)$ is the number of uncrawled changes source $w$ has accumulated by time $t$, and $C : \mathbb{Z}^+ \to \mathbb{R}^+$ is a *penalty function*, to be chosen later, that assigns a cost to every possible number of uncrawled changes. Note that $N_w(t)$ implicitly depends on the most recent time $w$ was crawled as well as on change sequence $ChSeq$, so the expectation is taken both over possible change sequences *and* possible crawl sequences $CrSeq$ generatable by $\pi$. Minimizing staleness means finding

$$\pi^* = \operatorname*{argmin}_{\pi \in \Pi} J^\pi,$$
(2)

subject to bandwidth constraints, where $\Pi$ is a suitably chosen policy class.

Choosing $C(n)$ that is efficient to optimize *and* induces "well-behaving" policies is of utmost importance. Cho & Garcia-Molina (2003a) and Azar et al. (2018) have consid-

ered Equation 1 with $C(n) = \mathbb{1}_{n>0}$, i.e., imposing a penalty if a source had any changes independently of their number, with $\Pi = \{CrSeq_w \sim Poisson(\rho_w) \text{ for all } w \in W | \vec{\rho}_w\}$, i.e., policies that crawl each source according to a Poisson process with a source-specific rate $\rho_w$. Under a bandwidth constraint only, they find the optimal policy in this class very efficiently, in $O(|W| \log(|W|))$ time, but this solution assigns $\rho_w = 0$ for many sources (Azar et al., 2018; Cho & Garcia-Molina, 2003a). In practice, this may not be acceptable, as it leaves the tracker stale w.r.t. these sources forever and therefore raises a question: why does the tracker monitor these sources at all?

In this paper, we propose and analyze the following penalty:

$$C(n) = H(n) = \sum_{i=1}^{n} \frac{1}{n} \text{ if } n > 0, \text{ and } 0 \text{ if } n = 0 \quad (3)$$

$H(n)$ for $n > 0$ is known as the $n$-th harmonic number and has several desirable properties as staleness penalty:

*It is strictly monotonically increasing.* Thus, it penalizes the tracker for every change that happened at a source since the previous crawl, not just the first one as in (Cho & Garcia-Molina, 2003a).

*It is discrete-concave, providing diminishing penalties.* This reflects the intuition that while all undownloaded changes at a source matter, the first one matters most, as it marks the transition from freshness to staleness.

*"Good" policies w.r.t. this objective don't starve any source:*

**Proposition 1.** *Under $C(n) = H(n)$, any policy that crawls each source at a fixed Poisson rate $\rho_w$ and assigns $\rho_w > 0$ to each $w$ with $\mu_w, \Delta_w > 0$ is strictly preferable to any such policy that assigns $\rho_w = 0$ to any such source.*

*Proof.* See the supplement. This is a direct consequence of Proposition 2: any policy $\pi$ with $\rho_w = 0$ has $J^\pi = \infty$. ∎

$C(n) = H(n)$ *allows for efficiently finding optimal policies under practical policy classes.* Indeed, $C(n) = H(n)$ isn't the only penalty function satisfying the above properties. For instance, $C(n) = n^d$ for $0 < d < 1$ and $C(n) = \log_d(1 + n)$ for $d > 1$ behave similarly. However, optimizing these alternatives turns out much less efficient.

## 4. Optimization under known change process

We now derive procedures for optimizing Equation 1 with $C(n) = H(n)$ (Equation 3) under the bandwidth constraint for sources with incomplete and complete change observations, assuming that we know the change process parameters $\vec{\Delta}$ exactly. In Section 5 we will lift the known-parameters assumption. We assume $\vec{\mu}, \vec{\Delta} > 0$, because sources that are unimportant or never change don't need to be crawled.

### 4.1. Case of incomplete change observations

When the tracker's only way to find out about changes at a source is crawling it, we consider the class of randomized memoryless policies that sample crawl times for each source according to a Poisson process with parameter $\rho_w$:

$$\Pi^- = \{CrSeq_w \sim Poisson(\rho_w) \, \forall w \in W^- | \vec{\rho} \geq 0\} \quad (4)$$

This policy class reflects the intuition that, since each source changes according to a Poisson process, i.e., roughly periodically, it should also be crawled roughly periodically. In fact, as Azar et al. (2018) show, any $\pi \in \Pi^-$ can be derandomized into a deterministic policy that is approximately periodic for each $w$. Since every $\pi \in \Pi^-$ is fully determined by the corresponding vector $\vec{\rho}$, we can easily express a bandwidth constraint on $\pi \in \Pi^-$ as $\sum_{w \in W^-} \rho_w = R$.

To optimize over $\Pi^-$, we first express the cost function from Equation 1 in terms of $\Pi^-$'s policy parameters $\vec{\rho} \geq 0$:

**Proposition 2.** *For $\pi \in \Pi^-$, $J^\pi$ from Eq. 1 is equivalent to*

$$J^\pi = - \sum_{w \in W^-} \mu_w \ln \left( \frac{\rho_w}{\Delta_w + \rho_w} \right) \quad (5)$$

*Proof.* See the Supplement. Note that $J^\pi = \infty$ if $\rho = 0$ for any $w \in W^-$. The proof proceeds via a series of algebraic manipulations and relies on properties of Poisson processes, particularly their memorylessness. ∎

Thus, finding $\pi^* \in \Pi^-$ can be formalized as follows:

**Problem 1.** *[Finding $\pi^* \in \Pi^-$]*

INPUT: *bandwidth $R > 0$; positive importance and change rate vectors $\vec{\mu}, \vec{\Delta} > 0$.*

OUTPUT: *Crawl rates $\vec{\rho} = (\rho_w)_{w \in W^-}$ maximizing*

$$\overline{J}^\pi = -J^\pi = \sum_{w \in W^-} \mu_w \ln \left( \frac{\rho_w}{\Delta_w + \rho_w} \right) \quad (6)$$

*subject to*

$$\sum_{w \in W^-} \rho_w = R, \rho_w \geq 0 \text{ for all } w \in W^-.$$

The next result readily identifies the optimal solution to this problem:

**Proposition 3.** *For $\vec{\mu}, \vec{\Delta} > 0$, policy $\pi^* \in \Pi^-$ parameterized by $\vec{\rho}^* > 0$ that satisfies*

$$\begin{cases} \rho_w = \frac{-\Delta_w + \sqrt{\Delta_w^2 + \frac{4\mu_w \Delta_w}{\lambda}}}{2}, & \text{for all } w \in W^- \\ \sum_{w \in W^-} \rho_w = R \end{cases} \quad (7)$$

*is unique, minimizes the harmonic penalty $J^\pi$ in Equation 3, and is therefore optimal in $\Pi^-$.*

*Proof.* See the Supplement. The main insight is that for any $\vec{\mu}, \vec{\Delta} > 0$ the Lagrange multiplier method, which gives

rise to Equation system 7, identifies the only maximizer of $\overline{J}^\pi = -J^\pi$ (Equation 6) in the region $\vec{\rho} > 0$, which must therefore correspond to $\pi^* \in \Pi^-$. Crucially, that solution always has $\lambda > 0$. ∎

Equation system 7 is non-linear, but the r.h.s. of equations involving $\lambda$ monotonically decreases in $\lambda > 0$, so, e.g., bisection search (Burden & Faires, 1985) on $\lambda > 0$ can find $\vec{\rho}^*$ as in Algorithm 1.

---

**Algorithm 1:** LAMBDACRAWL-INCOMLOBS: finding the optimal crawl scheduling policy $\pi^* \in \Pi^-$ under incomplete change observations (Problem 1)

**Input:** $R \geq 0$ – bandwidth;
$\vec{\mu} > 0, \vec{\Delta} > 0$ – importance and change rates;
$\epsilon > 0$ – desired precision on $\lambda$
1 ; **Output:** $\vec{\rho}$ – vector of crawl rates for each source.
2 lower_bound_$\lambda \hookleftarrow \frac{|W^-|^2 \min_{w \in W^-}\{\Delta_w\} \min_{w \in W^-}\{\mu_w\}}{|W^-| \max_{w \in W^-}\{\Delta_w\}R + R^2}$
3 upper_bound_$\lambda \hookleftarrow \frac{|W^-|^2 \max_{w \in W^-}\{\Delta_w\} \max_{w \in W^-}\{\mu_w\}}{|W^-| \min_{w \in W^-}\{\Delta_w\}R + R^2}$
4 $\lambda \hookleftarrow$ BisectionSearch(lower_bound_$\lambda$, upper_bound_$\lambda$, $\epsilon$)
5 // see, e.g., Burden & Faires (1985)
6 **foreach** $w \in W^-$ **do** $\rho_w \hookleftarrow \frac{-\Delta_w + \sqrt{\Delta_w^2 + \frac{4\mu_w \Delta_w}{\lambda}}}{2}$
7 Return $\vec{\rho}$

---

**Proposition 4.** LAMBDACRAWL-INCOMLOBS *(Algorithm 1) finds an arbitrarily close approximation to Problem 1's optimal solution in time* $O(\log_2(\frac{upper\_bound\_\lambda - lower\_bound\_\lambda}{\epsilon})|W^-|)$.

*Proof.* See the Supplement. The key step is showing that the solution $\lambda$ is in [lower_bound_$\lambda$, upper_bound_$\lambda$]. ∎

### 4.2. Case of complete change observations

When the tracker receives an observation every time a source changes, the policy class $\Pi^-$ in Equation 4 is clearly suboptimal, because it completely ignores change signals when deciding when to crawl a source. At the same time, crawling every source on every change signal is unviable, because the total change rate of all sources $\sum_{w \in W} \Delta_w$ can easily exceed bandwidth $R$. These extremes suggest a policy class whose members trigger crawls for only a fraction of the observations, dictated by a source-specific probability $p_w$:

$$\Pi^o = \{\text{for all } w \in W^o, \text{ whenever change observation } o_w$$
$$\text{arrives, crawl } w \text{ with probability } p_w \,|\, 0 \leq \vec{p} \leq 1\}$$

As with $\Pi^-$, to find $\pi^* \in \Pi^o$ we first express $J^\pi$ from Equation 1 in terms of $\Pi^o$'s policy parameters $\vec{p} = (p_w)_{w \in W^o}$:

**Proposition 5.** *For $\pi \in \Pi^o$, $J^\pi$ from Eq. 1 is equivalent to*

$$J^\pi = -\sum_{w \in W^o} \mu_w \ln(p_w) \qquad (8)$$

*if $\vec{p} > 0$ and $J^\pi = \infty$ if $p_w = 0$ for any $w \in W^o$.*

*Proof.* See the Supplement. The key insight is that under any $\pi \in \Pi^o$, the number of $w$'s uncrawled changes at time $t$ is geometrically distributed with parameter $p_w$. ∎

Under any $\pi \in \Pi^o$, the crawl rate $\rho_w$ of any source $w$ is related to its change rate $\Delta_w$: every time $w$ changes we get an observation and crawl $w$ with probability $p_w$. Thus, $\rho_w = p_w \Delta_w$. However, since $p_w$ is a probability, we must have $0 \leq p_w \leq 1$. Also, bandwidth $R > \sum_{w \in W^o} \Delta_w$ isn't sensible, because with complete change observations the tracker doesn't benefit from more crawls than there are changes. Thus, we frame finding $\pi^* \in \Pi^o$ as follows:

**Problem 2.** *[Finding $\pi^* \in \Pi^o$]*

INPUT: *bandwidth $R$ s.t. $0 < R \leq \sum_{w \in W^o} \Delta_w$; importance and change rate vectors $\vec{\mu}, \vec{\Delta} > 0$.*

OUTPUT: *Crawl probabilities $\vec{p} = (p_w)_{w \in W^o}$ maximizing*

$$\overline{J}^\pi = -J^\pi = \sum_{w \in W^o} \mu_w \ln(p_w) \qquad (9)$$

*subject to*

$$\sum_{w \in W^o} p_w \Delta_w = R, \;\; 0 \leq p_w \leq 1 \text{ for all } w \in W^o$$

Solving Problem 2 requires non-linear optimization under inequality constraints, which, in general, could take time exponential in the constraint number. Our main result in this subsection is that one can find the optimal solution to Problem 2 in polynomial time in the number of constraints.

We begin by inspecting solutions to the relaxation of Problem 2 that ignores the inequality constraints:

**Proposition 6.** *The optimal solution $\vec{\hat{p}}^*$ to the relaxation of Problem 2 that ignores inequality constraints is unique and assigns*

$$\hat{p}_w^* = \frac{R\mu_w}{\Delta_w \sum_{w' \in W^o} \mu_{w'}} \text{for all } w \in W^o \qquad (10)$$

*Proof.* See the Supplement. The proof applies the method of Lagrange multipliers to this relaxation. ∎

Equation 10 indicates that the relaxation's solution never has $p_w \leq 0$, but it may indeed violate the constraints $p_w \leq 1$. The main difficulty of inequality-constrained optimization is exactly in determining the subset of inequality constraints that are *active* under the optimal solution, i.e., in our case, finding all sources $w$ for which the optimal $\vec{p}^*$ has $p_w^* = 1$.

Our algorithm LAMBDACRAWL-COMPLOBS (Algorithm 2) finds them iteratively. In each iteration (lines 4-14), we consider only the sources that haven't yet been proved to activate their $p_w \leq 1$ constraint under $\vec{p}^*$ (lines 3,12). For them, we solve a relaxation of Problem 2 that *ignores* these

constraints, using Proposition 6 (lines 5-6), and check if this solution $\vec{\hat{p}}^*$ happens to violate any constraints (line 9). Our key insight is that any source $w$ that activates or violates its constraint under Proposition 6's solution, i.e., has $\hat{p}_w^* \geq 1$, must necessarily have $p_w^* = 1$. As we prove in the Supplement, this follows from the strict concavity of $\overline{J}^\pi$ and the convexity of the optimization region given by the constraints. Whenever we can prove source $w$ to activate its constraint in this way, we set $p_w^* = 1$, adjust the overall bandwidth constraint for the remaining sources to $R_{rem} = R - p_w^* \Delta_w = R - \Delta_w$, and remove $w$ from further consideration (lines 10-12). Eventually, our problem gets reduced to a (possibly empty) set of sources for which Proposition 6's solution doesn't violate any constraints under the remaining bandwidth (lines 15-16). Since Proposition 6's solution is optimal in this case, the overall algorithm is optimal as well.

---

**Algorithm 2:** LAMBDACRAWL-COMPLOBS: finding the optimal crawl scheduling policy $\pi^* \in \Pi^o$ under complete change observations (Problem 2)

---

1  LAMBDACRAWL-COMPLOBS:

 **Input:** $\vec{\mu}, \vec{\Delta}$ – importance and change rate vectors
2    $R$ s.t. $0 \leq R \leq \sum_{w \in W} \Delta_w$ – bandwidth;
 **Output:** $\vec{p}^*$ – vector specifying optimal per-page crawl probabilities upon receiving a change observation.
3  $W_{rem} \leftarrow W^o$      // remaining sources to consider
4  **while** $W_{rem}^o \neq \emptyset$ **do**
5   **foreach** $w \in W_{rem}^o$ **do**
6     $\hat{p}_w^* \leftarrow \frac{R\mu_w}{\Delta_w \sum_{w' \in W_{rem}^o} \mu_{w'}}$ for all $w \in W_{rem}^o$
7    $ViolationDetected \leftarrow False$
8   **foreach** $w \in W_{rem}^o$ **do**
9     **if** $\hat{p}_w^* \geq 1$ **then**
10      $p_w^* \leftarrow 1$
11      $R \leftarrow R - \Delta_w$ // reduce remaining bandwidth
12      $W_{rem}^o \leftarrow W_{rem}^o \setminus \{w\}$  // ignore $w$ onwards
13      $ViolationDetected = True$
14   **if** $ViolationDetected == False$ **then** break
15   **foreach** $w \in W_{rem}^o$ **do**
16     $p_w^* \leftarrow \hat{p}_w^*$
17  Return $\vec{p}^* = (p_w^*)_{w \in W^o}$

---

**Proposition 7.** LAMBDACRAWL-COMPLOBS *returns the optimal solution to Problem 2 and runs in time* $O(|W^o|^2)$.

*Proof.* See the Supplement. The proof formalizes the above intuitions and critically relies on the concavity of $\overline{J}^\pi$. ∎

In practice, the running time bound of $O(|W^o|^2)$ is very loose. Each iteration usually discovers several active constraints, and many sources don't activate their constraint under $\vec{p}^*$, so the computation time is close to linear in $|W^o|$.

## 4.3. Crawl scheduling under mixed observability

In practice, trackers have to simultaneously handle sources $W^-$ for which complete change observations are available and sources $W^o$ for which they aren't at the same time, under a common bandwidth constraint $R$. How should we optimize scheduling mixed-observability scenarios?

Consider the policy class that combines $\Pi^-$ and $\Pi^o$:

$$\Pi^\ominus = \begin{cases} \{CrSeq_w \sim Poisson(\rho_w) \text{ for all } w \in W^- | \vec{\rho}\}, \\ \{\text{for all } w \in W^o, \text{ whenever change observation} \\ o_w \text{ arrives, crawl } w \text{ with probability } p_w | \vec{p}\} \end{cases}$$
(11)

For $\pi \in \Pi^\ominus$, Propositions 2 and 5 imply that $J^\pi$ from Equation 1 is equivalent to

$$J^\pi = - \sum_{w \in W^-} \mu_w \ln \left( \frac{\rho_w}{\Delta_w + \rho_w} \right) - \sum_{w \in W^o} \mu_w \ln (p_w)$$
(12)

Optimization over $\pi \in \Pi^\ominus$ can be stated as follows:
**Problem 3.** *[Finding $\pi^* \in \Pi^\ominus$]*

INPUT: *bandwidth $R > 0$; importance and change rate vectors $\vec{\mu}, \vec{\Delta} > 0$.*

OUTPUT: *Crawl rates $\vec{\rho} = (\rho_w)_{w \in W^-}$ and crawl probabilities $\vec{p} = (p_w)_{w \in W^o}$ maximizing*

$$\overline{J}^\pi = -J^\pi = \sum_{w \in W^-} \mu_w \ln \left( \frac{\rho_w}{\Delta_w + \rho_w} \right) + \sum_{w \in W^o} \mu_w \ln (p_w)$$
(13)

*subject to*

$$\sum_{w \in W^-} \rho_w + \sum_{w \in W^o} p_w \Delta_w = R,$$

$$\rho_w > 0 \text{ for all } w \in W^-, \ \ 0 < p_w \leq 1 \text{ for all } w \in W^o$$

The optimization objective (Equation 13) is strictly concave as a sum of concave functions over the region described by the constraints, and therefore has a unique maximizer. To find it efficiently, we observe that solving Problem 3 amounts to deciding how to split the global bandwidth constraint $R$ into an allotment $R^o$ for sources with complete change observations and $R^- = R - R^o$ for the rest. For any candidate split, LAMBDACRAWL-COMPLOBS and LAMBDACRAWL-INCOMLOBS give us the reward-maximizing policy parameters $\vec{p}^*(R^o)$ and $\vec{\rho}^*(R^-)$, respectively, and Equation 13 then tells us the overall value $\overline{J}^*(R^o, R^-)$ of that split. We also know that for the optimal split, $R^{o^*} \in [0, \min\{R, \sum_{w \in W_o} \Delta_w\}]$, as described in Problem 2 and discussion immediately before it. Thus, we can find Problem 3's maximizer to any desired precision using a method such as Golden-section search (Kiefer, 1953) on $R^o$. LAMBDACRAWL (Algorithm 3) implements this idea, where SPLIT-EVAL-$\overline{J}^*$ (line 7) evaluates $\overline{J}^*(R^o, R^-)$ and OptMaxSearch denotes an optimal search method.

**Proposition 8.** LAMBDACRAWL *(Algorithm 3) finds an*

*arbtirarily close approximation to Problem 3's optimal solution using $O(\log(\frac{R}{\epsilon}))$ calls to* LAMBDACRAWL-INCOMLOBS *and* LAMBDACRAWL-COMPLOBS.

*Proof.* This follows directly from the optimality of LAMBDACRAWL-INCOMLOBS and LAMBDACRAWL-COMPLOBS (Propositions 3 and 7), as well as of OptMaxSearch such as Golden section, which makes $O(\log(\frac{R}{\epsilon}))$ iterations. ∎

---

**Algorithm 3:** LAMBDACRAWL: finding optimal mixed-observability policy $\pi^* \in \Pi^{\ominus}$ (Problem 3)

**Input:** $R > 0$ – bandwidth;
  $\vec{\mu} > 0, \vec{\Delta} > 0$ – importance and change rates;
  $\epsilon^{\text{no-obs}}, \epsilon > 0$ – desired precisions
**Output:** $\vec{\rho}^*, \vec{p}^*$ – crawl rates and probabilities for sources without and with complete change observations.
1 $R^o_{min} \hookleftarrow 0$
2 $R^o_{max} \hookleftarrow \min\{R, \sum_{w \in W_o} \Delta_w\}$
3 $\vec{\rho}^*, \vec{p}^* \hookleftarrow OptMaxSearch(\text{Split-Eval-}\overline{J}^*, R^o_{min}, R^o_{max}, \epsilon)$
4 // E.g., Golden section search (Kiefer, 1953)
5 Return $\vec{\rho}^*, \vec{p}^*$
6
7 SPLIT-EVAL-$\overline{J}^*$:
**Input:** $R^o$ – bandwidth for sources with complete change observations, $R, \vec{\mu}, \vec{\Delta}, \epsilon^{\text{no-obs}}$
**Output:** $\overline{J}^*$ (Equation 13) for the given split
8 $\vec{\rho} \hookleftarrow$ LAMBDACRAWL-INCOMLOBS$(R - R^o, \vec{\mu}_{W^-}, \vec{\Delta}_{W^-}, \epsilon^{\text{no-obs}})$
9 $\vec{p} \hookleftarrow$ LAMBDACRAWL-COMPLOBS$(R^o, \vec{\mu}_{W^o}, \vec{\Delta}_{W^o})$
10 Return $\sum_{w \in W^-} \mu_w \ln\left(\frac{\rho_w}{\Delta_w + \rho_w}\right) + \sum_{w \in W^o} \mu_w \ln(p_w)$

---

## 5. Reinforcement learning for scheduling

Although staleness minimization is efficient under the known-model assumption, in reality change rates are usually unavailable and vary with time, requiring constant re-learning. As it turns out, their estimation can be done with only minor changes to LAMBDACRAWL that turn it into a type of model-based reinforcement learning (RL) algorithm.

Suppose for a given source $w$ the tracker has observed a sequence of binary change indicator variables $z_1, z_2, \ldots, z_U$, where $t_0, \ldots, t_U$ are times when the observations arrived, and, for $1 \leq j \leq U$, $z_j = 1$ iff the source changed compared to time $t_{j-1}$ *at least* once. Consider two cases:

**Incomplete change observations for $w$.** Crawling the source is the only way for the tracker to find out about any changes: after each crawl at times $t_1, \ldots, t_U$, the tracker checks for changes compared to $w$'s previous crawl and records $z_{t_j} = 1$ or $z_{t_j} = 0$ accordingly. There may be more than one change since the previous crawl, but the tracker cannot determine this. Denoting $a_{t_j} = t_j - t_{j-1}, j \geq 1$, Cho & Garcia-Molina (2003b) show that $\hat{\Delta}$ that solves

$$\sum_{j:z_{t_j}=1} \frac{a_{t_j}}{e^{a_{t_j}\Delta} - 1} - \sum_{j:z_{t_j}=0} a_{t_j} = 0, \quad (14)$$

is a maximum-likelihood estimator of $\Delta$ for the given source. The l.h.s. of the equation is monotonically decreasing in $\Delta$, so $\hat{\Delta}$ can be efficiently found numerically. This estimator is consistent under mild conditions (Cho & Garcia-Molina, 2003b), e.g., if the sequence $\{a_{t_j}\}_{j=1}^{\infty}$ doesn't converge to 0, i.e., if the observations are spaced apart.

**Complete change observations for $w$.** In this case, for all $j$, $z_{t_j} = 1$; a change observation arrives only when there is a change and indicates *exactly one* change. This is a standard setting for estimating Poisson process intensity, and a consistent MLE estimator $\hat{\Delta}$ is simply the time-averaged number of observed changes (Taylor & Karlin, 1998):

$$\Delta = \frac{U + 1}{t_U}, \quad (15)$$

LAMBDALEARNANDCRAWL, a model-based RL version of LAMBDACRAWL that uses these estimators to learn model parameters simultaneously with scheduling is presented in Algorithm 4. It operates in epochs of length $T_{epoch}$ time units each (lines 3-13). At the start of each epoch $n$, it calls LAMBDACRAWL (Algorithm 3) on the available $\vec{\Delta}_{n-1}$ change rate estimates to produce a policy $(\vec{\rho}^*_n, \vec{p}^*_n)$ optimal with respect to them (line 4). Executing this policy during the current epoch, for the time period of $T_{epoch}$, and recording the observations extends the observation history (lines 7-8). (Note though that for sources $w \in W^o$, the observations don't depend on the policy.) It then re-estimates change rates using a suffix of the augmented observation history (lines 10-13). Under mild assumptions, LAMBDALEARNANDCRAWL converges to the optimal policy:

**Proposition 9.** LAMBDALEARNANDCRAWL *(Algorithm 4) converges in probability to the optimal policy under the true change rates $\vec{\Delta}$, i.e., $\lim_{N_{epochs} \to \infty}(\vec{\rho}_{N_{epochs}}, \vec{p}_{N_{epochs}}) = (\vec{\rho}^*, \vec{p}^*)$, if $\vec{\Delta}$ is stationary and $S(n)$, the length of the history's training suffix, satisfies $S(N_{epoch}) = length(obs\_hist)$.*

*Proof.* See the Supplement. It follows from the consistency and positivity of the change rate estimates, as well as LAMBDACRAWL's optimality ∎

Using LAMBDALEARNANDCRAWL in practice requires attention to several aspects:
*Stationarity of $\vec{\Delta}$.* Source change rates may vary with time, so the length of history suffix for estimating $\vec{\Delta}$ should typically be shorter than the entire available history.

*Singularities of $\vec{\Delta}$ estimators.* Given a limited observation history, the MLE in Equation 14 can produce $\hat{\Delta} = \infty$ if all crawls detected a change (the r.h.s. is 0). Similarly, Equation 15 can produce $\hat{\Delta}_w = 0$ if no observations about

---

**Algorithm 4:** LAMBDALEARNANDCRAWL: finding optimal crawl scheduling policy $\pi^* \in \Pi^\ominus$ (Problem 3) under initially unknown change model

---

**Input:** $R > 0$ – bandwidth;

$\vec{\mu} > 0, \vec{\Delta}_0 > 0$ – importance and initial change rate guesses

$\epsilon^{\text{no-obs}}, \epsilon > 0$ – desired precisions

$T_{epoch} > 0$ – duration of an epoch

$N_{epochs} > 0$ – number of epochs

$S(n)$ – for each epoch $n$, observation history suffix length for learning $\vec{\Delta}$ in that epoch

1 // $obs\_hist[S(n)]$ is $S(n)$-length observation history suffix

2 $obs\_hist \leftarrow ()$

3 **foreach** $1 \le n \le N_{epochs}$ **do**

4    $\vec{\rho}_n^*, \vec{p}_n^* \leftarrow$ LAMBDACRAWL$(R, \vec{\mu}, \vec{\Delta}_{n-1}, \epsilon^{\text{no-obs}}, \epsilon)$

5    // $\vec{Z}_{new}$ holds observations for all sources from start to

6    // end of epoch $n$. Execute policy $(\vec{\rho}_n^*, \vec{p}_n^*)$ to get it

7    $\vec{Z}_{new} \leftarrow ExecuteAndObserve(\vec{\rho}_n^*, \vec{p}_n^*, T_{epoch})$

8    $Append(obs\_hist, \vec{Z}_{new})$

9    // Learn new $\vec{\Delta}$ estimates using Eqs. 14 and 15

10    **foreach** $w \in W^-$ **do**

11      $\hat{\Delta}_{n_w} \leftarrow Solve\left( \sum\limits_{j:z_{j_w}=1} \frac{a_j}{e^{a_j\Delta}-1} + \frac{0.5}{e^{0.5\Delta}-1} - \sum\limits_{j:z_{j_w}=0} a_j - 0.5 = 0, obs\_hist[S(n)] \right)$

12    **foreach** $w \in W^o$ **do**

13      $\hat{\Delta}_{n_w} \leftarrow Solve\left( \frac{U_{S(n)}+0.5}{S(n)+0.5}, obs\_hist[S(n)] \right)$

---

$w$ arrived during a given period. To avoid them without affecting estimation consistency, we use smoothing to add imaginary observation intervals of length 0.5 to Equation 14 and imaginary 0.5 observation to Equation 15 (lines 11,13).

The sheer number of parameters LAMBDALEARNAND-CRAWL needs to learn raises a question: can we employ an approximation with fewer parameters to learn? The following result suggests one such approximation:

**Proposition 10.** *Suppose the tracker's set of sources $W^-$ is such that for some constant $c > 0$, $\frac{\mu_w}{\Delta_w} = c$ for all $w \in W^-$. Then minimizing harmonic penalty under incomplete change observations (Problem 1) has $\rho_w^* = \frac{\mu_w R^-}{\sum_{w' \in W^-} \mu_{w'}}$.*

*Proof.* See the Supplement. The proof proceeds by plugging in $\Delta_w = \frac{1}{c}\mu_w$ into Equation system 7. ∎

In essence, the proposition states that if the importance-to-change-rate ratio is constant across all sources, then each source's crawl rate is independent of its change rate, and even the ratio constant itself. This greatly simplifies crawl scheduling, because for sources $w \in W^-$, we don't need to learn any change rates, provided that the constant-ratio assumption holds at least approximately. Moreover, if LAMBDACRAWL-INCOMLOBS (Algorithm 1)

is replaced by assigning $\rho_w^*, w \in W^-$ as in Proposition 10, an $O(|W^-|)$ operation, which improves the computational efficiency of LAMBDACRAWL and LAMBDALEARNAND-CRAWL in its own right. Our evaluation (Section 7) explores the quality of this approximation empirically.

## 6. Related work

**Scheduling for Posting, Polling, and Maintenance.** Besides monitoring information sources, mathematically related settings arise in smart broadcasting in social networks (Karimi et al., 2016; Zarezade et al., 2017; Wang et al., 2017; Upadhyay et al., 2018), personalized teaching (Upadhyay et al., 2018), database synchronization (Gal & Eckstein, 2001), job scheduling (Glazebrook & Mitchell, 2002), and scheduling maintenance service to machines (Anily et al., 1998; Bar-Noy et al., 1998; Glazebrook et al., 2005). In the context of web crawling (see Olston & Najork (2010) for an overview), the closest works are (Cho & Garcia-Molina, 2003a), (Wolf et al., 2002), (Pandey & Olston, 2005), and (Azar et al., 2018). Like (Cho & Garcia-Molina, 2003a) and (Azar et al., 2018), we use Lagrange multipliers as part of optimization, and adopt the Poisson change model of (Cho & Garcia-Molina, 2003a) and many works since. Our contributions differ from prior art in several ways: (1) optimization objectives (see the next subsection) and policy guarantees; (2) special crawl scheduling under complete change observations, both separately from and jointly with the commonly studied setting where changes are detectable only at crawl time; (3) learning model parameters in a principled way during crawling.

**Optimization objectives.** Our objective falls in the class of convex separable resource allocation problems (Ibaraki & Katoh, 1988). So do most other related objectives: binary freshness/staleness (Azar et al., 2018; Cho & Garcia-Molina, 2003a), age (Cho & Garcia-Molina, 2000a), and embarrassment (Wolf et al., 2002). The latter is implemented via specially constructed importance scores (Wolf et al., 2002), so our algorithms can be used for it too. Other separable objectives include information longevity (Olston & Pandey, 2008). In contrast, Pandey & Olston (2005) focus on an objective that depends on user behavior and cannot be separated into contributions from individual sources. While intuitively appealing, their measure can be optimized only via many approximations (Pandey & Olston, 2005), and the algorithm for it is ultimately heuristic.

**Acquiring model parameters.** Importance can be defined and quickly determined from information readily available to search engines, e.g., page relevance to queries (Wolf et al., 2002), query-independent popularity such as PageRank (Page et al., 1998), and other features (Pandey & Olston, 2005; Radinsky & Bennett, 2013). Learning change rates is more delicate. Change rate estimators we use are due to (Cho & Garcia-Molina, 2003b); our contribution in this regard is integrating them into crawl scheduling while providing theoretical guarantees, as well as identifying conditions

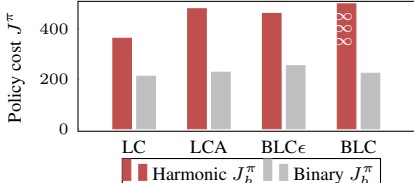

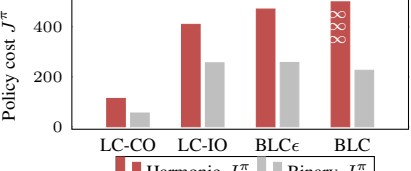

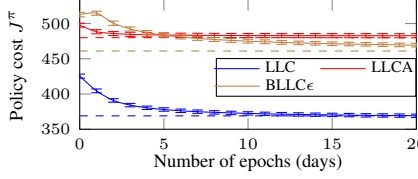

*Figure 1.* Performance w.r.t. harmonic ($J_h^\pi$) and binary ($J_b^\pi$) policy costs. **Lower bars = better policies.** *LC* is robust to both, but *BLCϵ* & *BLC* (Azar et al., 2018) aren't: *LC* ($J_h^\pi$-optimal) beats *BLCϵ* by 22% and *BLC* ($J_h^\pi = \infty$) in $J_h^\pi$, but *BLC* ($J_h^\pi$-optimal) and *BLCϵ* don't beat *LC/LCA* in $J_b^\pi$. *CC* ($J_h^\pi = 1791, J_b^\pi = 891$) and *UC* ($J_h^\pi = 1237, J_b^\pi = 636$) did poorly (omitted from the plot).

*Figure 2.* Benefit of using complete change observations. Here we use only the 13% of our dataset's URLs that provide them (via sitemaps). LC-ComplObs (*LC-CO*, Alg. 2) heeds these signals while LC-IncomplObs (*LC-IO*, Alg. 1) and others ignore them. As a result, *LC-CO*'s policy cost both w.r.t. $J_h^\pi$ and $J_b^\pi$ is at least $4\times$ (!) lower than the other algorithms'.

*Figure 3.* Convergence of the RL-based *LLC*, *LLCA*, and *BLLCϵ* initialized with random page change rate estimates. Dashed lines show asymptotic policy costs $J_h^\pi$; plots have confidence intervals. *LLC* converges much faster than *BLLCϵ*, *LLCA* even more so. *LLCA*'s asymptotic policy is worse than *LLC*'s but comparable to *BLLCϵ*'s, especially w.r.t. binary cost $J_b^\pi$ (Fig. 1).

when estimation can be side-stepped using an approximation (Prop. 10). While many works adopted the homogeneous Poisson change process (Cho & Garcia-Molina, 2000b; Cho & Ntoulas, 2002; Cho & Garcia-Molina, 2003a;b; 2000a; Wolf et al., 2002; Azar et al., 2018), its non-homogeneous variant (Gal & Eckstein, 2001), quasi-deterministic (Wolf et al., 2002), and general marked temporal point process (Upadhyay et al., 2018) change models were also considered. Change models can also be inferred via generalization using source co-location (Cho & Ntoulas, 2002) or similarity (Radinsky & Bennett, 2013).

**RL.** Our setting could be viewed as a *restless* multi-armed bandit (MAB) (Whittle, 1988), a MAB type that allows an arm to change its reward/cost distribution without being pulled. However, no known restless MAB class allows arms to *incur* a cost/reward without being pulled, as in our setting. This distinction makes existing MAB analysis such as (Immorlica & Kleinberg, 2018) inapplicable to our model. RL with events and policies obeying general marked temporal point processes was studied in (Upadhyay et al., 2018). However, it relies on DNNs and as a result doesn't provide guarantees of convergence, optimality, other policy properties, or a mechanism for imposing strict constraints on bandwidth, and is far more expensive computationally.

## 7. Empirical evaluation

Our results support three claims: (1) LAMBDACRAWL's harmonic staleness cost $J_h^\pi$ (Eqs. 1, 3) is a more robust objective than the binary cost $J_b^\pi$ widely studied previously (e.g., (Azar et al., 2018; Cho & Garcia-Molina, 2003a; Wolf et al., 2002)). Optimizing the former yields policies that are also near-optimal w.r.t. the latter, while the converse is not true (Figure 1). (2) Utilizing complete remote observations as LAMBDACRAWL-COMPLOBS does when they are available makes a very big difference in policy cost (Figure 2). (3) Using Prop. 10's approximation instead of LAMBDACRAWL-INCOMLOBS for $w \in W^-$ reduces performance w.r.t. $J_h^\pi$ but speeds up RL convergence if source

change rates are initially unknown (Figure 3). At the same time, this approximation only weakly affects performance w.r.t. binary cost $J_h^\pi$ (Figure 1). These factors and algorithm simplicity make this approximation a useful tradeoff in practice.

The experiments used web page change and importance data collected by crawling 18,532,326 URLs daily for 14 weeks. We compared LAMBDACRAWL (labeled *LC* in the figures), LAMBDACRAWLAPPROX (*LCA*, *LC* with Prop. 10's approximation), and their RL variants *LLC* (Alg. 4) and *LLCA* to *BinaryLambdaCrawl* (*BLC*) (Azar et al., 2018), the state-of-the-art optimal algorithm for minimizing binary cost $J_b^\pi$. Since *BLC* may crawl-starve sources and hence get $J_h^\pi = \infty$ (see Fig. 1), we also use our own variant of it, *BLCϵ*, with the non-starvation guarantee, and its RL version *BLLCϵ*. Finally, we also use *ChangeRateCrawl* (*CC*) (Cho & Garcia-Molina, 2003a; Wolf et al., 2002) and *UniformCrawl* (*UC*) (Cho & Garcia-Molina, 2000b; Olston & Pandey, 2008) heuristics. **Please see Supplement, Sec. 9 for the algorithm descriptions and further details of the experiment setup.**

## 8. Conclusion

We have introduced a new objective and a suite of highly efficient algorithms for it to address the freshness crawl scheduling problem faced by many services from search engines to databases. In particular, we have presented LAMBDALEARNANDCRAWL, which integrates model parameter learning with scheduling optimization. To give convergence speed guarantees for this kind of approaches in the future, we intend to look at multi-armed bandit (MAB) models similar to Immorlica & Kleinberg (2018). Their current fundamental distinction from our setting is the reward mechanism: a MAB yields a reward only when an arm is pulled, not continuously as in freshness optimization. Nonetheless, we believe that a new bandit model will provide practically and theoretically important insights into freshness crawl scheduling.

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

# SUPPLEMENT

## 9. Details of the Experiments and Additional Plots

### 9.1. Dataset, Implementation, Hardware

For the dataset, we crawled $18,532,326$ URLs daily over 14 weeks to estimate their change rates reliably using Equations 14 and 15. Some of the URL crawls on some days failed for reasons ranging from crawler's internal errors to the URL host being temporarily unavailable, so many URLs were crawled fewer than $14 \cdot 7 = 98$ times. At the same time, some URLs were crawled more often as part of the crawler's other workloads.

These URLs are data sources for the knowledge base of a major virtual assistant. The knowledge base uses special information extractors to get important information out of these pages. To determine if a page changed, we ran the same information extractors on it every time we crawled it and considered the page as changed if the extracted information changed.

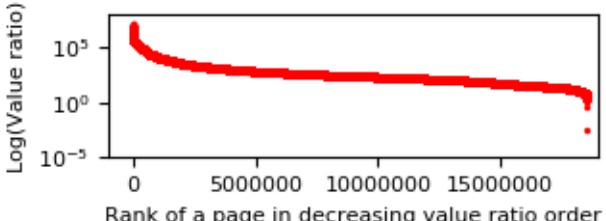

*Figure 4.* Ranking of 18,532,326 URLs in our dataset by their $\frac{\mu_w}{\Delta_w}$ ratio.

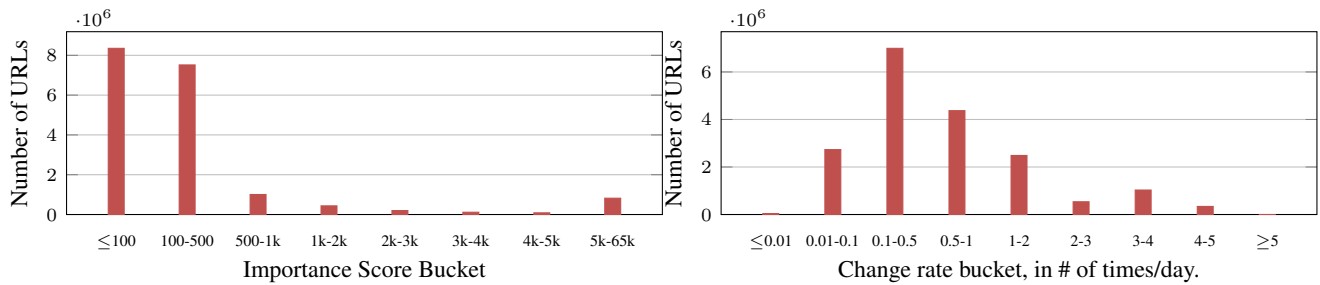

*Figure 5.* Importance score histogram for URLs in our dataset. The distribution has a big skew, with most pages having importance less than 1000.

*Figure 6.* (Poisson) change rate distribution for URLs in our dataset. Most URLs change once in a few days.

13% of the URLs in the dataset had complete change observations that we obtained by frequently crawling reliable sitemaps associated with these URLs.

We set the URL importance scores $\mu_w$ to values defined by the production crawler based on PageRank and popularity.

Proposition 10 suggests that the performance gap between LAMBDACRAWL and LAMBDACRAWLAPPROX depends on the distribution of the ratios $\frac{\mu_w}{\Delta_w}$ across the set of sources $W$: if they are all equal, the policy cost of LAMBDACRAWL and LAMBDACRAWLAPPROX should be the same. As Figure 4 shows, these ratios are similar across much of our dataset, but clearly non-uniform in the head and tail of the ranking. Figures 5 and 6 show the dataset's importance and change rate distributions.

In each run of an algorithm, **the bandwidth constraint was set to 20% of the number of pages used in that run.**

All algorithms used in the experiments were implemented in Python (*the code is submitted with the paper and will be open-sourced if the paper is accepted*) and run on a Windows 10 laptop with 16GB RAM and an Intel quad-core 2.11GHz i7-8650U CPU.

**The dataset and the code will be released if the paper is accepted.**

### 9.2. Evaluation metrics

To evaluate the algorithms' performance, we used two metrics:

- **The harmonic policy cost** $J_h^\pi$ as in Equation 1 with $C(n)$ as in Equation 3.

- **The binary policy cost** $J_b^\pi$ as in Equation 1 with $C(n)$ defined as

$$C(n) = \mathbb{1}_{n>0}$$

.

This policy cost objective was studied in several works including (Cho & Garcia-Molina, 2003a; Azar et al., 2018). Some prior research considered its finite-horizon (Wolf et al., 2002) and discrete-time versions (Azar et al., 2018). Note that some of these papers formulated their objective as *maximizing freshness*, whereby the agent is *rewarded* for each time unit when the number of accumulated changes at a source is 0. Maximizing this objective means minimizing binary staleness (although the two aren't necessarily negations of each other!) Thus, the two are equivalent and we don't distinguish between them in the paper.

Actually using $J_b^\pi$ to evaluate policy requires deriving its parameterization in terms of policy $\pi$'s crawl rates $\vec{\rho}$ and crawl probabilities $\vec{\pi}$, analogously to Propositions 2 and 5 for the harmonic cost $J_h^\pi$. By following the steps in the proofs of these propositions, we derived the following expressions for $J_b^\pi$:

$$J_b^\pi = \sum_{w \in W^-} \frac{\mu_w \Delta_w}{\Delta_w + \rho_w} \quad \text{for pages } w \in W^- \tag{16}$$

$$J_b^\pi = \sum_{w \in W^o} \mu_w (1 - p_w) \quad \text{for pages } w \in W^o \tag{17}$$

Note that Equation 16 is similar to the equation in Azar et al. (2018) for evaluating the *freshness reward* of policy $\pi$.

**For each experiment, we report the values of $J_h^\pi$ and $J_b^\pi$ normalized by the number of URLs used in that experiment.**

### 9.3. Algorithms

In the experiments description in Section 7, we refer to the following algorithms used in the empirical evaluation:

LAMBDACRAWL (*LC*), as in Algorithm 3.

LAMBDACRAWL-COMPLOBS (**LC-ComplObs**, *LC-CO*), as in Algorithm 2.

LAMBDACRAWL-INCOMLOBS (**LC-IncomlObs**, *LC-IO*), as in Algorithm 1.

*BinaryLambdaCrawl* (*BLC*) is the name we give to the state-of-the-art, optimal algorithm proposed by Azar et al. (2018) for minimizing binary staleness $J_b^\pi$. *BLC* is one of two major baselines for LAMBDACRAWL in our experiments.

The difference between *BLC*'s objective $J_b^\pi$ and *LC*'s objective $J_h^\pi$ is crucial in practice, because minimizing binary staleness $J_b^\pi$ generally yields $\vec{\rho}^*$ with $\rho_w^* = 0$ for many sources even if they have $\Delta_w > 0$. This effectively tells the tracker to ignore changes to these sources — an unacceptable strategy in real applications. Indeed, harmonic penalty (Equation 1) assigns $J_h^\pi = \infty$ to such strategies.

***BinaryLambdaCrawl***($\epsilon$) (***BLC$\epsilon$***) Vanilla *BLC*'s lack of non-starvation guarantees makes comparing it to LAMBDACRAWL in terms of $J_h^\pi$ uninsightful, because for *BLC*, $J_h^\pi$ is usually $\infty$. To address this issue, and simultaneously make *BLC* more practical, we modified *BLC* to enforce the non-starvation guarantee. The resulting algorithm is *BLC$\epsilon$*.

Namely, *BLC$\epsilon$* accepts a parameter $\epsilon \in [0, 1]$. Initially, it operates exactly like *BLC* to find $\vec{\rho^*}$ optimal w.r.t. the binary cost $J_b^\pi$. Then it finds all sources $w$ for which

$$\rho_w^* < \epsilon R/|W|,$$

sets $\rho_w^* = \epsilon R/|W|$ for each of them, and re-solves the problem over the remaining sources and bandwidth again using *BLC*. Thus, *BLC$\epsilon$* uniformly distributes a small fraction of the bandwidth to sources that would otherwise get no or little crawl rate allocated to it.

The original, optimal *BLC* can be viewed as *BLC*(0). Any $\epsilon > 0$ results in suboptimality w.r.t. $J_b^\pi$ while ensuring that $J_h^\pi < \infty$. For the experiments, we did a parameter sweep to determine $\epsilon$ that resulted in *BLC$\epsilon$*'s best performance w.r.t. LAMBDACRAWL's objective $J_h^\pi$. **The best value we found for our dataset, and used in all the experiments, is $\epsilon = 0.4$.**

***UniformCrawl*** (***UC***) (Cho & Garcia-Molina, 2000b; Olston & Pandey, 2008) is a heuristic that assigns an equal crawl rate to all sources:

$$\rho_w = R/|W|.$$

In spite of its simplicity, in our experiments it outperforms *ChangeRateCrawl*, as predicted by Cho & Garcia-Molina (2003a) (see the caption of Figure 1).

***ChangeRateCrawl*** (***CC***) is the name we give to another heuristic proposed by Cho & Garcia-Molina (2003a) that sets

$$\rho_w = \frac{\Delta_w R}{\sum_{w' \in W} \Delta_{w'}},$$

thereby crawling sources at a rate proportional to their change rate. Cho & Garcia-Molina (2003a) pointed out that *ChangeRateCrawl* can be very suboptimal if the set $W$ includes sources that change frequently. This causes *ChangeRateCrawl* to over-commit crawl bandwidth to sources whose changes are near-impossible to keep up with, at the expense of almost ignoring the rest. Our experimental results agree with this observation — *ChangeRateCrawl* turned out to be the weakest-performing algorithm in our experiments.

LAMBDALEARNANDCRAWL (***LLC***), as in Algorithm 4.

LAMBDALEARNANDCRAWLAPPROX (***LLCA***), as in Algorithm 4 with calls to LAMBDACRAWL-INCOMLOBS in LAMBDACRAWL replaced by setting

$$\rho_w = \frac{\mu_w R}{\sum_{w' \in W^-} \mu_{w'}}.$$

per Proposition 10.

***BinaryLambdaLearnAndCrawl***($\epsilon$) (***BLLC$\epsilon$***), the reinforcement learning version of *BLC$\epsilon$* where *BLC$\epsilon$* replaces LAMBDACRAWL in Algorithm 4. This is also the natural RL adaptation of pure *BLC* (Azar et al., 2018), which otherwise would need a dedicated exploration parameter to ensure data gathering for sources that would get $\rho_w = 0$ under *BLC*'s (currently) optimal policy. As for *BLC$\epsilon$*, we used $\epsilon = 0.4$.

### 9.4. Experiment 1 (Figure 1)

The goal of this experiment was to assess the harmonic objective $J_h^\pi$ that we proposed and the binary objective $J_b^\pi$ widely studied in previous works in terms of robustness: how well do policies optimal w.r.t. one of them behave w.r.t. the other, and vice versa?

In this experiment, we assumed known change rates. To obtain them, we inferred them with estimators in Equations 14 and 15 from our entire 14-week crawl data for 18.5M URLs, and used the resulting estimates as ground truth. Policies were evaluated by plugging in these change rates and policy parameters into the equations in Propositions 2 and 5 and into Equations 16 and 17.

As Figure 1 shows, the harmonic penalty $J_h^\pi$ we propose is a more flexible choice of objective. LAMBDACRAWL, optimal w.r.t. it, significantly outperforms $BLC\epsilon$ and $BLC$ w.r.t. it, and, which is more surprising, even the approximate LAMBDACRAWLAPPROX performs at par with $BLC\epsilon$ according to this objective. Moreover, LAMBDACRAWL manages to perform as well as $BLC$ on the binary objective $J_b^\pi$ for which $BLC$ is optimal. That is, no matter which objective we trust, optimizing for $J_h^\pi$ yields excellent results.

As a side note, the *UniformCrawl* and *ChangeRateCrawl* heuristics were outperformed by a large margin by the above methods w.r.t. both objectives.

### 9.5. Experiment 2 (Figure 2)

One of our contributions is a mechanism for taking advantage of complete remote change observations (Algorithm 2). $BLC\epsilon$, *BLC*, *UniformCrawl*, and *ChangeRateCrawl* don't have it, and treat all pages as if the only change observations for them came from crawling. While only 13% of web pages in our dataset have an (approximately) complete observation history, can LAMBDACRAWL's and LAMBDACRAWLAPPROX's advantage on them explain the performance gap in experiment 1?

Figure 2 indicates that these URLs are indeed responsible for a significant fraction of LAMBDACRAWL's advantage. In this experiment, we focused only on the above URLs. Like in the previous experiment, we assumed perfect model knowledge using the previously obtained change rate estimates and estimated policy performance using Propositions 2, 5 and Equations 16 and 17. Treating these URLs as complete-observation URLs, as LAMBDACRAWL-COMPLOBS does, resulted in nearly 5-fold reduction in policy cost, compared to treating these URLs under the conventional change observation model.

Although this gives LAMBDACRAWL an edge over previously proposed techniques, note that even when LAMBDACRAWL treats these URLs conventionally (denoted by the LAMBDACRAWL-INCOMLOBS (*LC-IO*) plot in Figure 2), still noticeably outperforms the other algorithms w.r.t. $J_h^\pi$ while holding its own against them w.r.t. $J_b^\pi$.

### 9.6. Experiment 3 (Figure 3)

Last but not least, we analyze the reinforcement learning variants of LAMBDACRAWL and LAMBDACRAWLAPPROX. Our motivation for the approximation in Proposition 10 was reducing the number of parameters LAMBDALEARNANDCRAWL has to learn in order to speed up convergence. In this experiment, we explore the tradeoff between the resulting gain in learning speed and the concomitant loss in solution quality.

The evaluation was done in a series of simulated episodes, each episode being executed on a randomly chosen 100,000-URL subsample of the 18.5M URLs (in each episode, the subsample was the same for all three algorithms.) In each episode, we simulated a 21-day run of *LLC*, *LLCA*, and *BLLC$\epsilon$* starting with random change rates. One epoch (see Algorithm 4) corresponded to 1 day. That is, every (simulated) day each of these algorithms re-estimated the change parameters using the simulated observation data (the simulated data wasn't shared among the algorithms). The simulated data was generated by sampling page changes using the ground truth change rates obtained in previous experiments and sampling page crawls from each algorithm's current policy. At the end of each day, each algorithm reoptimized its policy for the new estimates, and this policy was evaluated using the aforementioned equations. We performed 20 such episodes and averaged the policy values on each day.

Figure 3 demonstrates that LAMBDALEARNANDCRAWLAPPROX indeed converges quicker than the other algorithms, with its asymptotic performance comparable to *BLLC$\epsilon$*'s but falling short of LAMBDALEARNANDCRAWL's. *BLLC$\epsilon$*'s convergence was the slowest. It could potentially be improved by choosing a larger $\epsilon$ at the beginning and gradually "cooling" it. LAMBDALEARNANDCRAWL's advantage, besides convergence speed and asymptotic performance, is that it converges quickly without such parameter tuning. LAMBDALEARNANDCRAWLAPPROX can also be a viable alternative based on its simplicity and convergence rate.

## 10. Proofs

**PROOF OF PROPOSITION 1.** This is a direct consequence of Proposition 2, which implies that any policy $\pi$ with $\rho_w = 0$ for any source $w$ with $\mu_w, \Delta_w > 0$ has $J^\pi = \infty$, whereas any $\pi$ with $rho_w > 0$ for every $\mu_w, \Delta_w > 0$ has $J^\pi = \infty$    ∎.

**PROOF OF PROPOSITION 2.**

First we rearrange Equation 1 as follows, dropping the distributions under the expectation and using $W$ instead of $W^-$ throughout the proof to make the notation less cumbersome:

$$J^\pi = \lim_{T \to \infty} \mathbb{E}_{\substack{CrSeq \sim \pi, \\ ChSeq \sim P(\vec{\Delta})}} \left[ \frac{1}{T} \int_0^T \left( \sum_{w \in W} \mu_w C(N_w(t)) \right) dt \right]$$

$$= \lim_{T \to \infty} \frac{1}{T} \int_0^T \left( \sum_{w \in W} \mu_w \mathbb{E}\left[C(N_w(t))\right] \right) dt$$

Then we use the definition of expectation, chain rule of probabilities, and variable $t_{prev}$ to denote the time when source $w$ was last crawled before time $t$ (although $t_{prev}$ is specific to each source $w$, for clarity of notation we make this implicit):

$$J^\pi = \lim_{T \to \infty} \frac{1}{T} \int_0^T \left( \sum_{w \in W} \mu_w \mathbb{E}[C(N_w(t))] \right) dt$$

$$= \lim_{T \to \infty} \frac{1}{T} \int_0^T \left( \sum_{w \in W} \mu_w \sum_{m=0}^{\infty} \left( C(m) \cdot \mathbb{P}[N_w(t) = m] \right) \right) dt$$

$$= \lim_{T \to \infty} \frac{1}{T} \int_0^T \left( \sum_{w \in W} \mu_w \sum_{m=0}^{\infty} \left( C(m) \int_0^t \mathbb{P}[N_w(t) = m \mid t - t_{prev} = T'] \, \mathbb{P}[t - t_{prev} = T']dT' \right) \right) dt$$

Now using the fact that our policy is a set of Poisson processes with parameters $\rho_w$ and page changes are governed by another set of Poisson processes with parameters $\Delta_w$, we can plug in appropriate expressions for the probabilities:

$$J^\pi = \lim_{T \to \infty} \frac{1}{T} \int_0^T \left( \sum_{w \in W} \mu_w \sum_{m=0}^{\infty} \left( C(m) \int_0^t \left( \frac{(T'\Delta_w)^m e^{-T'\Delta_w}}{m!} \right) \left( \rho e^{-\rho_w T'} \right) dT' \right) \right) dt$$

$$= \lim_{T \to \infty} \frac{1}{T} \int_0^T \left( \sum_{w \in W} \mu_w \sum_{m=0}^{\infty} \left( \frac{C(m)\rho_w \Delta_w^m}{m!} \int_0^t T'^m e^{-(\rho_w + \Delta_w)T'} dT' \right) \right) dt$$

We do a variable substitution $u = (\rho_w + \Delta_w)T'$, so $T' = \frac{u}{\Delta_w + \rho_w}$ and $dT' = \frac{du}{\Delta_w + \rho_w}$:

$$J^\pi = \lim_{T \to \infty} \frac{1}{T} \int_0^T \left( \sum_{w \in W} \mu_w \sum_{m=0}^\infty \left( \frac{C(m) \rho_w \Delta_w^m}{m!} \int_0^t \left( \frac{u}{\Delta_w + \rho_w} \right)^m e^{-u} \frac{du}{\Delta_w + \rho_w} \right) \right) dt$$

$$= \lim_{T \to \infty} \frac{1}{T} \int_0^T \left( \sum_{w \in W} \mu_w \sum_{m=0}^\infty \left( \frac{C(m)}{m!} \left( \frac{\rho_w}{\Delta_w + \rho_w} \right) \left( \frac{\Delta_w}{\Delta_w + \rho_w} \right)^m \int_0^t u^m e^{-u} du \right) \right) dt$$

Consider $F(m,t) = \int_0^t u^m e^{-u} du$. By definition of gamma functions, $F(m,t) = \Gamma(m+1) - \Gamma(m+1, t) = m! - \Gamma(m+1, t)$. Recalling that $C(m) = H(m)$ for $m > 0$ and $C(0) = 0$ (Equation 3), we get

$$J^\pi = \lim_{T \to \infty} \frac{1}{T} \int_0^T \left( \sum_{w \in W} \mu_w \sum_{m=1}^\infty \left( \frac{H(m)}{m!} \left( \frac{\rho_w}{\Delta_w + \rho_w} \right) \left( \frac{\Delta_w}{\Delta_w + \rho_w} \right)^m (m! - \Gamma(m+1, t)) \right) \right) dt$$

$$= \lim_{T \to \infty} \frac{1}{T} \left( \sum_{w \in W} \mu_w \left( \frac{\rho_w}{\Delta_w + \rho_w} \right) \int_0^T \left[ \sum_{m=1}^\infty \left( H(m) \left( \frac{\Delta_w}{\Delta_w + \rho_w} \right)^m \right) - \sum_{m=1}^\infty \left( \frac{H(m)}{m!} \left( \frac{\Delta_w}{\Delta_w + \rho_w} \right)^m \Gamma(m+1, t) \right) \right] dt \right)$$

Now consider for each $w \in W$ functions $G(T) = \int_0^T \sum_{m=1}^\infty \left( H(m) \left( \frac{\Delta_w}{\Delta_w + \rho_w} \right)^m \right) dt$, $R(t) = \int_0^T \sum_{m=1}^\infty \left( \frac{H(m)}{m!} \left( \frac{\Delta_w}{\Delta_w + \rho_w} \right)^m \Gamma(m+1, t) \right) dt$, and $\lim_{T \to \infty} \frac{1}{T}(G(T) - R(T))$ so that

$$J^\pi = \sum_{w \in W} \left( \mu_w \left( \frac{\rho_w}{\Delta_w + \rho_w} \right) \left( \lim_{T \to \infty} \frac{1}{T}(G(T) - R(T)) \right) \right) \tag{18}$$

Thus, if $\lim_{T \to \infty} \frac{1}{T}(G(T) - R(T))$ exists, is finite, and we can compute it, we can compute $J^\pi$ as well. Focusing on $G(T)$ and recalling that $\sum_{m=1}^\infty H(m) x^m = -\frac{\ln(1-x)}{1-x}$ for $|x| < 1$, we see that $G(T) = -\frac{\ln(\frac{\rho_w}{\Delta_w + \rho_w})}{\frac{\rho_w}{\Delta_w + \rho_w}} T$, so $\lim_{T \to \infty} \frac{G(T)}{T} = -\frac{\ln(\frac{\rho_w}{\Delta_w + \rho_w})}{\frac{\rho_w}{\Delta_w + \rho_w}}$ exists. Focusing on $R(T)$, we see that since $m! < \Gamma(m+1, t)$ for any $t > 0$ and since $R(T)$ is an integral of a non-negative function, we have $0 \le R(T) \le G(T) < \infty$, so $\lim_{T \to \infty} \frac{R(T)}{T}$ exists as well. Therefore, we can write $\lim_{T \to \infty} \frac{1}{T}(G(T) - R(T)) = \lim_{T \to \infty} \frac{G(T)}{T} - \lim_{T \to \infty} \frac{R(T)}{T}$.

We already know $\lim_{T \to \infty} \frac{G(T)}{T}$. To evaluate $\lim_{T \to \infty} \frac{R(T)}{T}$, we upper-bound $R(T)$. Again using the fact that it is an integral of a non-negative function, we have

$$R(T) = \int_0^T \sum_{m=1}^\infty \left( \frac{H(m)}{m!} \left( \frac{\Delta_w}{\Delta_w + \rho_w} \right)^m \Gamma(m+1, t) \right) dt$$

$$< \int_0^\infty \sum_{m=1}^\infty \left( \frac{H(m)}{m!} \left( \frac{\Delta_w}{\Delta_w + \rho_w} \right)^m \Gamma(m+1, t) \right) dt$$

$$= \sum_{m=1}^\infty \left( \frac{H(m)}{m!} \left( \frac{\Delta_w}{\Delta_w + \rho_w} \right)^m \int_0^\infty \Gamma(m+1, t) dt \right)$$

$$= \sum_{m=1}^\infty \left( \frac{H(m)}{m!} \left( \frac{\Delta_w}{\Delta_w + \rho_w} \right)^m \Gamma(m+2) \right)$$

$$= \sum_{m=1}^\infty \left( (m+1) H(m) \left( \frac{\Delta_w}{\Delta_w + \rho_w} \right)^m \right)$$

We have used the fact that $\int_0^\infty \Gamma(m+1, t) dt = \Gamma(m+2) = (m+1)!$. The series $\sum_{m=1}^\infty (m+1) H(m) x^m$ converges for

$|x| < 1$ to some limit $L > 0$, so we have $\lim_{T\to\infty} \frac{R(T)}{T} \leq \lim_{T\to\infty} \frac{L}{T} = 0$ and $\lim_{T\to\infty} \frac{1}{T}(G(T) - R(T)) = -\frac{\ln(\frac{\rho_w}{\Delta_w + \rho_w})}{\frac{\rho_w}{\Delta_w + \rho_w}}$.
Plugging this back into Equation 10, we get

$$J^\pi = -\sum_{w \in W} \mu_w \ln\left(\frac{\rho_w}{\Delta_w + \rho_w}\right)$$

∎

**PROOF OF PROPOSITION 3.** The high-level idea is to apply the method of Lagrange multipliers to Problem 1's relaxation *without* inequality constraints $\vec{\rho} \geq 0$ and show that (a) only one local maximum of this relaxation is within the region given by $\vec{\rho} \geq 0$ – the one satisfying Equation system 7 – and (b) solutions that touch the boundary of this region, i.e., have $\rho_w = 0$ for any $w \in W^-$, are suboptimal. In fact, part (b) follows immediately from Proposition 1, so we only need to solve the relaxation and show part (a).

To apply the method of Lagrange multipliers to the relaxation, we set $f(\vec{\rho}) = \overline{J}^\pi = \sum_{w \in W^-} \mu_w \ln\left(\frac{\rho_w}{\Delta_w + \rho_w}\right)$ and $g(\vec{\rho}) = \sum_{w \in W^-} \rho_w - R$. We need to solve

$$\begin{cases} \nabla f(\vec{\rho}) = \lambda \nabla g(\vec{\rho}) \\ g(\vec{\rho}) = 0. \end{cases}$$

For any $w \in W^-$, we have $\frac{\partial g}{\partial \rho_w} = 1$ and $\frac{\partial f}{\partial \rho_w} = \mu_w \frac{\Delta_w + \rho_w}{\rho_w} \frac{\Delta_w}{(\Delta_w + \rho_w)^2} = \frac{\mu_w \Delta_w}{\Delta_w \rho_w + \rho_w^2}$, so the above system of equations turns into

$$\begin{cases} \frac{\mu_w \Delta_w}{\Delta_w \rho_w + \rho_w^2} = \lambda, & \text{for all } w \in W^- \\ \sum_{w \in W^-} \rho_w = R \end{cases}$$

and therefore

$$\begin{cases} \lambda \rho_w^2 + \lambda \Delta_w \rho_w - \mu_w \Delta_w = 0, & \text{for all } w \in W^- \\ \sum_{w \in W^-} \rho_w = R \end{cases}$$

Solving each quadratic equation separately, we get

$$\begin{cases} \rho_w = \frac{-\Delta_w \pm \sqrt{\Delta_w^2 + \frac{4\mu_w \Delta_w}{\lambda}}}{2}, & \text{for all } w \in W^- \\ \sum_{w \in W^-} \rho_w = R \end{cases}$$

This gives all potential solutions to the relaxation of Problem 1. Now consider the inequality constraints $\vec{\rho} \geq 0$ omitted so far. Observe that any real solution to the above system that has $\rho_w = \frac{-\Delta_w - \sqrt{\Delta_w^2 + \frac{4\mu_w \Delta_w}{\lambda}}}{2}$ for any $w \in W^-$ implies $\rho_w < 0$ for $\mu_w, \Delta_w > 0$, which violates these constraints. Therefore, any solution to Problem 1 itself must satisfy

$$\begin{cases} \rho_w = \frac{-\Delta_w + \sqrt{\Delta_w^2 + \frac{4\mu_w \Delta_w}{\lambda}}}{2}, & \text{for all } w \in W^- \\ \sum_{w \in W^-} \rho_w = R \end{cases}$$

and have $\lambda \geq 0$ (otherwise $\vec{\rho} < 0$, again violating the inequality constraints). Although the first group of equations are non-linear, note that each $\rho_w(\lambda)$ is strictly monotone decreasing in $\lambda$ for $\lambda \geq 0$, so $\sum_{w \in W^-} \rho_w$ is strictly monotone decreasing too implying that $\sum_{w \in W^-} \rho_w(\lambda) = R$ has a unique solution in $\lambda$, and therefore there is a unique $\vec{\rho}$ satisfying the above system of equations. Thus, this $\vec{\rho}$ is the solution to Problem 1 and therefore corresponds to $\pi^* \in \Pi^-$. ∎

### PROOF OF PROPOSITION 4

We start by combining Equation system 7,

$$\begin{cases} \rho_w = \frac{-\Delta_w + \sqrt{\Delta_w^2 + \frac{4\mu_w \Delta_w}{\lambda}}}{2}, & \text{for all } w \in W^- \\ \sum_{w \in W^-} \rho_w = R, \end{cases} \tag{19}$$

into one equation:

$$\sum_{w \in W^-} \frac{-\Delta_w + \sqrt{\Delta_w^2 + \frac{4\mu_w \Delta_w}{\lambda}}}{2} = R$$

Bisection search is guaranteed to converge to *some* solution $\lambda$ of this equation as long as we initialize the search with lower_bound_$\lambda$, upper_bound_$\lambda$ s.t. $\lambda \in [\text{lower\_bound\_}\lambda, \text{upper\_bound\_}\lambda]$. However, all $\lambda_- \leq 0$ that solve Equation system 7 correspond to solutions that have $\vec{\rho} < 0$. At the same time, from the proof of Proposition 3 we know that there is exactly one $\lambda_+ > 0$ that solves Equation system 7, and it corresponds to the (unique) the optimal solution $\vec{\rho}^*$ of Problem 1. We want bisection search to find only this $\lambda_+ > 0$, so we want lower_bound_$\lambda$, upper_bound_$\lambda$ s.t. $\lambda_+ \in [\text{lower\_bound\_}\lambda, \text{upper\_bound\_}\lambda]$ *and* lower_bound_$\lambda$, upper_bound_$\lambda > 0$.

To find these bounds, we observe that the l.h.s. of the above equation is monotonically decreasing in $\lambda$ for $\lambda > 0$, so if we find any $\lambda_l > 0$ that guarantees

$$\sqrt{\Delta_w^2 + \frac{4\mu_w \Delta_w}{\lambda_l}} \geq \Delta_w + \frac{2R}{|W|} \text{ for all } w \in W^- ,$$

then we have

$$\sum_{w \in W^-} \frac{-\Delta_w + \sqrt{\Delta_w^2 + \frac{4\mu_w \Delta_w}{\lambda_l}}}{2} \geq R$$

and hence $\lambda_l \leq \lambda_+$. To find such $\lambda_l$, we perform a series of algebraic manipulations:

$$\sqrt{\Delta_w^2 + \frac{4\mu_w \Delta_w}{\lambda_l}} \geq \Delta_w + \frac{2R}{|W|} \text{ for all } w \in W^-, \lambda_l > 0$$

$$\Longleftrightarrow \Delta_w^2 + \frac{4\mu_w \Delta_w}{\lambda_l} \geq \left(\Delta_w + \frac{2R}{|W^-|}\right)^2 \text{ for all } w \in W^-, \lambda_l > 0$$

$$\Longleftrightarrow \frac{4\mu_w \Delta_w}{\lambda_l} \geq \frac{4\Delta_w R}{|W^-|} + 4\left(\frac{R}{|W^-|}\right)^2 \text{ for all } w \in W^-, \lambda_l > 0$$

$$\Longleftrightarrow \lambda_l \leq \frac{|W^-|^2 \mu_w \Delta_w}{|W^-|\Delta_w R + R^2} \text{ for all } w \in W^-, \lambda_l > 0$$

$$\Longleftarrow \lambda_l \leq \frac{|W^-|^2 \min_{w \in W^-}\{\mu_w\} \min_{w \in W^-}\{\Delta_w\}}{|W^-| \max_{w \in W^-}\{\Delta_w\}R + R^2}, \lambda_l > 0$$

Since the r.h.s. of this inequality is always positive, we set lower_bound_$\lambda = \frac{|W^-|^2 \min_{w \in W^-}\{\mu_w\} \min_{w \in W^-}\{\Delta_w\}}{|W^-| \max_{w \in W^-}\{\Delta_w\}R + R^2}$. An

analogous chain of reasoning shows that we can choose upper_bound_$\lambda = \frac{|W^-|^2 \max_{w \in W^-}\{\Delta_w\} \max_{w \in W^-}\{\mu_w\}}{|W^-| \min_{w \in W^-}\{\Delta_w\} R + R^2}$.

To establish a bound on the running time, we observe that each iteration of LAMBDACRAWL-INCOMLOBS involves evaluating $\sum_{w \in W^-} \rho_w$, which takes $O(|W^-|)$ time, so by the properties of bisection search the total running time is $O(\log_2(\frac{\text{upper\_bound\_}\lambda - \text{lower\_bound\_}\lambda}{\epsilon})|W^-|)$. ∎

**PROOF OF PROPOSITION 5** Let $Ch_w(t)$ denote the total number of changes that have happened at source $w$ in time interval $[0, t]$. As in the proof of Proposition 2, we start with rearranging the cost function in Equation 1, dropping the distributions under the expectation and using $W$ instead of $W^o$ throughout the proof to make the notation less cumbersome. Using the definition of expectation and chain rule of probabilities to get:

$$
\begin{aligned}
J^\pi &= \lim_{T \to \infty} \mathop{\mathbb{E}}_{\substack{CrSeq \sim \pi, \\ ChSeq \sim P(\vec{\Delta})}} \left[ \frac{1}{T} \int_0^T \left( \sum_{w \in W} \mu_w C(N_w(t)) \right) dt \right] \\
&= \lim_{T \to \infty} \frac{1}{T} \int_0^T \left( \sum_{w \in W} \mu_w \mathbb{E}\left[ C(N_w(t)) \right] \right) dt \\
&= \lim_{T \to \infty} \frac{1}{T} \int_0^T \left( \sum_{w \in W} \mu_w \sum_{m=0}^\infty \left( C(m) \cdot \mathbb{P}[N_w(t) = m] \right) \right) dt \\
&= \lim_{T \to \infty} \frac{1}{T} \int_0^T \left( \sum_{w \in W} \mu_w \sum_{c=0}^\infty \sum_{m=0}^\infty \left( C(m) \cdot \mathbb{P}[N_w(t) = m | Ch_w(t) = c] \mathbb{P}[Ch_w(t) = c] \right) \right) dt
\end{aligned}
$$

Since changes at every source $w$ are governed by a Poisson process with rate $\Delta_w$, $\mathbb{P}[Ch_w(t) = c] = \frac{e^{-\Delta_w t}(\Delta_w t)^c}{c!}$. Now, consider $\mathbb{P}[N_w(t) = m | Ch_w(t) = c]$. Recall that whenever source $w \in W^o$ changes, we find out about the change immediately; with probability $p_w$ the policy $\pi \in \Pi^o$ then crawls source $w$ straight away, and with probability $(1 - p_w)$ it waits to make this decision until we find out about $w$'s next change. Therefore, the only way we can have $N_w(t) = m$ is if our policy crawled source $w$ $m + 1$ changes ago *and* has *not* chosen to crawl $w$ after any of the $m$ changes that happened since, or it hasn't chosen to crawl $w$ since "the beginning of time". Thus, $N_w(t)$ is geometrically distributed, assuming that at least $m + 1$ changes actually happened at source $w$ in the time interval $[0, 1]$. Thus, we have

$$
\mathbb{P}[N_w(t) = m | Ch_w(t) = c] = \begin{cases} p_w(1 - p_w)^m, & \text{if } c \geq m + 1 \\ (1 - p_w)^m, & \text{if } c = m \\ 0 & \text{otherwise} \end{cases}
$$

Recalling that $C(m) = H(m)$ for $m > 0$ and $C(0) = 0$ (Equation 3) and putting everything together, we have

$$
\begin{aligned}
J^\pi &= \lim_{T \to \infty} \frac{1}{T} \int_0^T \left( \sum_{w \in W} \mu_w \sum_{c=0}^\infty \sum_{m=0}^\infty \left( C(m) \cdot \mathbb{P}[N_w(t) = m | Ch_w(t) = c] \mathbb{P}[Ch_w(t) = c] \right) \right) dt \\
&= \lim_{T \to \infty} \frac{1}{T} \int_0^T \left( \sum_{w \in W} \mu_w \sum_{c=1}^\infty \left( \frac{e^{-\Delta_w t}(\Delta_w t)^c}{c!} \right) \left( p_w \sum_{m=1}^{c-1} H(m)(1 - p_w)^m + H(c)(1 - p_w)^c \right) \right) dt \\
&= \lim_{T \to \infty} \frac{1}{T} \int_0^T \left( \sum_{w \in W} \mu_w \sum_{c=1}^\infty \left( \frac{e^{-\Delta_w t}(\Delta_w t)^c}{c!} \right) \left( p_w \sum_{m=1}^\infty H(m)(1 - p_w)^m - p_w \sum_{m=c}^\infty H(m)(1 - p_w)^m + H(c)(1 - p_w)^c \right) \right) dt
\end{aligned}
$$

Now consider

$$G(T) = \int_0^T \left( \sum_{w \in W} \mu_w \sum_{c=0}^{\infty} \left( \frac{e^{-\Delta_w t}(\Delta_w t)^c}{c!} \right) \left( p_w \sum_{m=0}^{\infty} C(m)(1-p_w)^m \right) \right) dt$$

$$R(T) = \int_0^T \left( \sum_{w \in W} \mu_w \sum_{c=0}^{\infty} \left( \frac{e^{-\Delta_w t}(\Delta_w t)^c}{c!} \right) \left( p_w \sum_{m=c}^{\infty} C(m)(1-p_w)^m \right) \right) dt$$

$$F(T) = \int_0^T \left( \sum_{w \in W} \mu_w \sum_{c=0}^{\infty} \left( \frac{e^{-\Delta_w t}(\Delta_w t)^c}{c!} \right) \left( C(c)(1-p_w)^c \right) \right) dt$$

Since

$$J^\pi = \lim_{T \to \infty} \frac{1}{T}(G(T) - R(T) + F(T)),$$

if we show that each of $\lim_{T \to \infty} \frac{G(T)}{T}$, $\lim_{T \to \infty} \frac{R(T)}{T}$, and $\lim_{T \to \infty} \frac{F(T)}{T}$ exists and manage to compute them, then we will know $J^\pi$. The rest of the proof focuses on computing these limits.

**Consider** $G(T)$. Note that $p_w \sum_{m=0}^{\infty} C(m)(1-p_w)^m = p_w \sum_{m=1}^{\infty} H(m)(1-p_w)^m$ doesn't depend on $c$. We can use the identity $\sum_{m=1}^{\infty} H(m)x^m = -\frac{\ln(1-x)}{1-x}$ for $|x| < 1$ to get $p_w \sum_{m=0}^{\infty} C(m)(1-p_w)^m = -\ln(p_w)$, so $G(T) = -\int_0^T \left( \sum_{w \in W} \mu_w \sum_{c=0}^{\infty} \left( \frac{e^{-\Delta_w t}(\Delta_w t)^c}{c!} \right) \ln(p_w) \right) dt$. To simplify $G(T)$ further, note that $\sum_{c=0}^{\infty} \left( \frac{e^{-\Delta_w t}(\Delta_w t)^c}{c!} \right)$ is just the probability of *any* number of changes occurring in time interval $[0, t]$ under a Poisson process, and therefore equals 1. Thus, $G(T) = -\int_0^T \left( \sum_{w \in W} \mu_w \ln(p_w) \right) dt = -T \sum_{w \in W} \mu_w \ln(p_w)$, so

$$\lim_{T \to \infty} \frac{G(T)}{T} = -\sum_{w \in W} \mu_w \ln(p_w).$$

**Consider** $R(T)$. Observe that $C(m) = H(m) < m$ for $m > 1$, so $p_w \sum_{m=c}^{\infty} C(m)(1-p_w)^m < p_w \sum_{m=c}^{\infty} m(1-p_w)^m = \frac{(1-p_w)^c(cp_w - p_w + 1)}{p_w}$. Because $(1 - p_w)^c$ decreases in $c$ much faster than $\frac{1}{c(cp_w - p_w + 1)}$ for any fixed $0 < p_w \le 1$, for any $w \in W$ there is a $c_w^*$ s.t. $p_w \sum_{m=c}^{\infty} C(m)(1-p_w)^m < \frac{(1-p_w)^c(cp_w - p_w + 1)}{p_w} < \frac{1}{c}$ for any $c > c_w^*$. Let $c^* = \max_{w \in W} c_w^*$. We can then upper-bound $R(T)$ as follows:

$$R(T) = \int_0^T \left( \sum_{w \in W} \mu_w \sum_{c=0}^{\infty} \left( \frac{e^{-\Delta_w t}(\Delta_w t)^c}{c!} \right) \left( p_w \sum_{m=c}^{\infty} C(m)(1-p_w)^m \right) \right) dt = R_1(T) + R_2(T),$$

where

$$R_1(T) = \int_0^T \left( \sum_{w \in W} \mu_w \sum_{c=0}^{c^*-1} \left( \frac{e^{-\Delta_w t}(\Delta_w t)^c}{c!} \right) \left( p_w \sum_{m=c}^{\infty} C(m)(1-p_w)^m \right) \right) dt$$

$$R_2(T) = \int_0^T \left( \sum_{w \in W} \mu_w \sum_{c=c^*}^{\infty} \left( \frac{e^{-\Delta_w t}(\Delta_w t)^c}{c!} \right) \left( p_w \sum_{m=c}^{\infty} C(m)(1-p_w)^m \right) \right) dt$$

Considering $R_1(T)$, recalling that $C(m) = H(m)$ for $m > 0$, we have $p_w \sum_{m=c}^{\infty} C(m)(1-p_w)^m \le p_w \sum_{m=1}^{\infty} C(m)(1-p_w)^m = -\ln(p_w)$, as shown previously. Also, for any $c$ we have $\int_0^T \frac{e^{-\Delta_w t}(\Delta_w t)^c}{c!} dt = \frac{\Gamma(c+1) - \Gamma(c+1, \Delta_w T)}{\Delta_w c!}$. Therefore,

$$R_1(T) = \int_0^T \left( \sum_{w \in W} \mu_w \sum_{c=0}^{c^*-1} \left( \frac{e^{-\Delta_w t}(\Delta_w t)^c}{c!} \right) \left( p_w \sum_{m=c}^\infty C(m)(1-p_w)^m \right) \right) dt$$

$$< - \sum_{w \in W} \mu_w \ln(p_w) \sum_{c=0}^{c^*-1} \frac{\Gamma(c+1) - \Gamma(c+1, \Delta_w T)}{\Delta_w c!}$$

Since $\lim_{T \to \infty} \frac{\Gamma(c+1, \Delta_w T)}{T} = 0$, $R_1(T)$ is therefore upper-bounded by a finite sum of terms that all go to 0 when divided by $T$ as $T \to \infty$. Since $R_1(T) \geq 0$ also holds, we have $\lim_{T \to \infty} \frac{R_1(T)}{T} = 0$.

Considering $R_2(T)$, we use our definition of $c^*$ to write

$$R_2(T) = \int_0^T \left( \sum_{w \in W} \mu_w \sum_{c=c^*}^\infty \left( \frac{e^{-\Delta_w t}(\Delta_w t)^c}{c!} \right) \left( p_w \sum_{m=c}^\infty C(m)(1-p_w)^m \right) \right) dt$$

$$< \int_0^T \left( \sum_{w \in W} \mu_w \sum_{c=c^*}^\infty \left( \frac{e^{-\Delta_w t}(\Delta_w t)^c}{c!} \right) \left( \frac{1}{c} \right) \right) dt = \int_0^T \left( \sum_{w \in W} \mu_w \sum_{c=c^*}^\infty \left( \frac{e^{-\Delta_w t}(\Delta_w t)^c}{(c+1)!} \right) \right) dt$$

$$\leq \int_0^T \left( \sum_{w \in W} \mu_w \sum_{c=0}^\infty \left( \frac{e^{-\Delta_w t}(\Delta_w t)^c}{(c+1)!} \right) \right) dt$$

$$= \int_0^T \left( \sum_{w \in W} \mu_w \left( \frac{1 - e^{-\Delta_w t}}{\Delta_w t} \right) \right) dt$$

$$= \sum_{w \in W} \mu_w \frac{\ln(\Delta_w T) + \Gamma(0, \Delta_w T) + \gamma}{\Delta_w},$$

where $\gamma$ is the Euler-Mascheroni constant. Since $\lim_{T \to \infty} \frac{\Gamma(0, \Delta_w T)}{T} = 0$, $R_2(T)$ is therefore upper-bounded by a finite sum of terms that all go to 0 when divided by $T$ as $T \to \infty$. Since $R_2(T) \geq 0$ also holds, we have $\lim_{T \to \infty} \frac{R_2(T)}{T} = 0$. Thus, we have

$$\lim_{T \to \infty} \frac{R(T)}{T} = \lim_{T \to \infty} \frac{R_1(T) + R_2(T)}{T} = \lim_{T \to \infty} \frac{R_1(T)}{T} + \lim_{T \to \infty} \frac{R_2(T)}{T} = 0$$

**Consider** $F(T)$. Observe that for a suitably chosen constant $s > 0$, $sR(T) > F(T)$. Therefore, since $\lim_{T \to \infty} \frac{R(T)}{T} = 0$, $\lim_{T \to \infty} \frac{F(T)}{T} = 0$ too.

We have thus shown that

$$J^\pi = \lim_{T \to \infty} \frac{G(T) - R(t) + F(T)}{T} = \lim_{T \to \infty} \frac{G(T)}{T} - \lim_{T \to \infty} \frac{R(T)}{T} + \lim_{T \to \infty} \frac{F(T)}{T} = - \sum_{w \in W} \mu_w \ln(p_w)$$

∎

**PROOF OF PROPOSITION 6.** Since under any $\vec{p} \geq 0$ crawl rates $\vec{\rho}$ are related to crawl probabilities via $\rho_w = p_w \Delta_w$, to apply the method of Lagrange multipliers to the relaxation of Problem 2 that takes into account only the bandwidth constraint we set $f(\vec{p}) = \overline{J}^\pi = \sum_{w \in W^\circ} \mu_w \ln(p_w)$ and $g(\vec{\rho}) = \sum_{w \in W^\circ} p_w \Delta_w - R$. We need to solve

$$\begin{cases} \nabla f(\vec{p}) = \lambda \nabla g(\vec{p}) \\ g(\vec{p}) = 0. \end{cases}$$

For any $w \in W^o$, we have $\frac{\partial g}{\partial p_w} = \Delta_w$ and $\frac{\partial f}{\partial p_w} = \frac{\mu_w}{p_w}$, so the above system of equations turns into

$$\begin{cases} \frac{\mu_w}{p_w} = \lambda \Delta_w, & \text{for all } w \in W^o \\ \sum_{w \in W^o} p_w \Delta_w = R \end{cases}$$

and therefore

$$\begin{cases} p_w = \frac{R \mu_w}{\Delta_w \sum_{w \in W^o} \mu_w} \text{for all } w \in W^o \\ \lambda = \frac{\sum_{w \in W^o} \mu_w}{R} \end{cases}$$

This is the only solution yielded by the method of Lagrange multipliers, so it is the unique maximizer of the relaxation. ∎

### PROOF OF PROPOSITION 7.

To establish LAMBDACRAWL-COMPLOBS's correctness, we first prove the following lemma, which establishes that any source $w$ that violates its $p_w \leq 1$ constraint in any iteration of LAMBDACRAWL-COMPLOBS must have $p_w^* = 1$:

**Lemma 1.** *Let $\vec{p}^*$ be the maximizer of Problem 2, and let $\vec{p}'^*$ be the maximizer of the relaxation Problem 2 with the same inputs but with inequality constraints ignored. Then any source $w$ that has $\vec{p}'^*$ has $p_w'^* > 1$, violating its inequality constraint, necessarily has $p_w^* = 1$.*

*Proof.* For convenience, we rewrite Problem 2 as an equivalent problem of maximizing $\overline{J}_{mod} = \sum_{w \in W^o} \mu_w \ln(\rho_w)$ under the constraints $\sum_{w \in W^o} \rho_w = R$ and $\rho_w \leq \Delta_w$ for every $w \in W^o$. Consider its relaxation with $\rho_w \leq \Delta_w$ for every $w \in W^o$ ignored. By the equivalent of Proposition 6 for this reformulation, $\vec{\rho}'^*$ is unique and must have $p_w'^* = \rho_w'^*/\Delta_w$ for each source $w$. The Lagrangian of $\mathcal{L}(\vec{\rho}, \lambda_0) := \overline{J}_{mod}(\vec{\rho}) - \lambda_0(\sum_{w \in W^o} \rho_w - R)$ must have $\nabla \mathcal{L}(\vec{\rho}'^*, \lambda_0'^*) = 0$ for the optimal solution $(\vec{\rho}'^*, \lambda_0'^*)$ (which, again, encodes the optimal $p_w'^*$ for the relaxation of the original formulation of Problem 2 via $p_w'^* = \rho_w'^*/\Delta_w$). From this we see that $\frac{\partial}{\partial \rho_u}\overline{J}_{mod} \restriction_{\rho_u'^*} = \lambda_0'^* = \frac{\partial}{\partial \rho_w}\overline{J}_{mod} \restriction_{\rho_w'^*}$, for any sources $u, w \in W^o$. Since $\frac{\partial}{\partial \rho_w}\overline{J}_{mod} = \frac{\mu_w}{\rho_w}$, this implies

$$\frac{\mu_u}{\rho_u'^*} = \frac{\mu_w}{\rho_w'^*} \quad \text{for all } u, w \in W^o. \tag{20}$$

Consider the slack-variable formulation of Problem 2 with slack variables $\{q_w\}_{w \in W^o}$. Inequality constraints in this formulation turn into $\rho_w = \Delta_w - q_w$ for $q_w \geq 0$. In this formulation, we now have

$$\mathcal{L}(\vec{\rho}, \lambda_0, \lambda_1, \dots, \lambda_{|W^o|}) := \mathcal{L}(\vec{\rho}, \lambda_0) - \sum_{w \in W^o} \lambda_w(\rho_w - \Delta_w).$$

where, by the Karush-Kuhn-Tucker conditions, $\lambda_w \geq 0$ for all $w$. By complementary slackness, $\lambda_w^* = 0$ for every $w \in W^o$ such that $q_w > 0$, i.e., for every $w$ that does *not* activate its inequality constraint, under the optimal solution $(\vec{\rho}^*, \lambda_0^*, \lambda_w^*, \dots, \lambda_{|W^o|}^*)$. This implies that $\frac{\partial}{\partial \rho_u}\overline{J}_{mod} \restriction_{\rho_u^*} -\lambda_u^* = \lambda_0^* = \frac{\partial}{\partial \rho_w}\overline{J}_{mod} \restriction_{\rho_w^*} -\lambda_w^*$, for any sources $u, w \in W^o$, i.e.,

$$\frac{\mu_u}{\rho_u'^*} - \lambda_u^* = \lambda_0^* = \frac{\mu_w}{\rho_w'^*} - \lambda_w^* \quad \text{for all } u, w \in W^o. \tag{21}$$

Now, suppose for contradiction that there is a source $u \in W^o$ that has $p_u'^* > 1$ but $p_u^* < 1$, implying that $\rho_u'^* > \Delta_u$ but $\rho_u^* < \Delta_u$. This, in turn, implies that (a) $\rho_u'^* > \rho_u^*$ and (b) $\lambda_u^* = 0$, since $u$ doesn't activate its inequality constraint under $\vec{\rho}^*$ (and hence under $\vec{p}^*$). Then, since $\sum_{v \in W^o} \rho_v^* = \sum_{v \in W^o} \rho_v'^* = R$, there must also exist some other source $w \neq u$ such that $\rho_w'^* < \rho_w^*$. For this source, $\lambda_w^* \geq 0$, so $\lambda_w^* \geq \lambda_u^*$.

Recall that $\overline{J}_{mod}$ is strictly concave, and its partial derivatives $\frac{\mu_v}{\rho_v}$ are monotone decreasing in every non-negative $\rho_v$. Together with $\rho'^*_u > \rho^*_u$, $\rho'^*_w < \rho^*_w$, $\lambda^*_w \geq \lambda^*_u$, and Equation 21, this implies

$$\frac{\mu_u}{\rho'^*_u} < \frac{\mu_u}{\rho^*_u} \leq \frac{\mu_w}{\rho^*_w} < \frac{\mu_w}{\rho'^*_w}$$

But this contradicts Equation 20, completing the proof of the lemma. ∎

The optimality of LAMBDACRAWL-COMPLOBS now follows by induction. Its every iteration except the last one identifies at least one constraint that is active under $\bar{p}^*$, by the above lemma, and thereby assigns an optimal $p^*_w$ to some sources, leaving optimal crawl probabilities for others to be found in subsequent iterations. The solution for the sources remaining in the final iteration, which does not violate any inequality constraints, is optimal by Proposition 6. Therefore, LAMBDACRAWL-COMPLOBS arrives at the optimal solution, and that solution is unique because Problem 2's maximization objective is concave as a sum of concave functions, and the optimization region is convex.

We note that the proof so far is similar to the proof of Lemma 3.3 from (Kolobov et al., 2019) for a different concave function $F$ under constraints of the form $x_1 + \ldots + x_k \leq c_k$, where $x_1, \ldots, x_k$ is a subset of $F$'s variables and $c_k$ is a constraint.

Since in each iteration LAMBDACRAWL-COMPLOBS removes at least one source from further consideration, it makes at most $|W^o|$ iterations. In each iteration it applies Proposition 6, which takes $O(|W^o|)$ time, yielding the overall time complexity of $O(|W^o|^2)$. ∎

**PROOF OF PROPOSITION 8.** See the paper. ∎

**PROOF OF PROPOSITION 9.** LAMBDALEARNANDCRAWL starts with strictly positive finite estimates $\vec{\Delta}_0$ of change rates. Since LAMBDACRAWL, which LAMBDALEARNANDCRAWL uses for determining crawl rates for the next epoch, is optimal to any desired precision (Proposition 8), it follows from Proposition 1 that it returns positive $\vec{\rho}^*_1, \vec{p}^*_1 > 0$, and $0 < \vec{\mu}, R < \infty$ guarantees that these crawl rates are also finite. In subsequent iterations, $\vec{\Delta}_n$ are estimated using Equations 14 and 15 with smoothing terms (lines 11 and 13 of Algorithm 4), and the smoothing terms ensure that the change rate estimates are finite and bounded away from 0: $0 < \delta_{min} \leq \vec{\Delta}_n < \infty$, where $\delta_{min}$ is implied by the aforementioned estimators and specific smoothing term values. This, along with finite positive $\vec{\mu}$ and $R$, ensures that $0 < \vec{\rho}^*_{n+1}, \vec{p}^*_{n+1} < \delta_{max}$. Hence, by induction, no source is ever starved, and no source is crawled infinitely frequently. This ensures, together with consistency of estimators from Equations 14 and 15, that if at least the last iteration $N_{epoch}$ uses the entire observation history, i.e. $S(N_{epoch}) = length(obs\_hist)$, then change rate estimates converge to the true change rates in probability: $\text{plim}_{N_{epochs} \to \infty} \vec{\Delta}_{N_{epochs}} = \vec{\Delta}$, as long as $\vec{\Delta}$ doesn't change with time. Optimality of LAMBDACRAWL then implies probabilistic convergence of $(\vec{\rho}^*_{N_{epochs}}, \vec{p}^*_{N_{epochs}})$ to $(\vec{\rho}^*, \vec{p}^*)$ as well. ∎

**PROOF OF PROPOSITION 10.** According to Equation system 7, the parameter vector $\vec{\rho}$ of the optimal $\pi^* \in \Pi^-$ that minimizes the expected harmonic penalty in the absence of remote change observations (Problem 1) satisfies

$$\begin{cases} \rho_w = \frac{-\Delta_w + \sqrt{\Delta_w^2 + \frac{4\mu_w \Delta_w}{\lambda}}}{2}, & \text{for all } w \in W^- \\ \sum_{w \in W^-} \rho_w = R \end{cases}$$

If $\frac{\mu_w}{\Delta_w} = c$ for all $w \in W^-$, $c > 0$, then we can express $\Delta_w = c' \mu_w$ where $c' = 1/c$ and plug it into the above equations to get

$$\rho_w = \frac{-\Delta_w + \sqrt{\Delta_w^2 + \frac{4\mu_w \Delta_w}{\lambda}}}{2}$$

$$= \frac{-c'\mu_w + \sqrt{(c'\mu_w)^2 + \frac{4c'\mu_w^2}{\lambda}}}{2}$$

$$= \frac{-c'\mu_w + \mu_w \sqrt{\frac{c'^2\lambda + 4c'}{\lambda}}}{2}$$

$$= \mu_w \left( \frac{-c' + \sqrt{\frac{c'^2\lambda + 4c'}{\lambda}}}{2} \right) \quad \text{for all } w \in W^- \tag{22}$$

Plugging this into the remaining equation from the above system, $\sum_{w \in W^-} \rho_w = R$, we get

$$\sum_{w \in W^-} \mu_w \left( \frac{-c' + \sqrt{\frac{c'^2\lambda + 4c'}{\lambda}}}{2} \right) = R \implies$$

$$\frac{-c' + \sqrt{\frac{c'^2\lambda + 4c'}{\lambda}}}{2} = \frac{R}{\sum_{w \in W^-} \mu_w} \implies$$

$$\frac{c'^2\lambda + 4c'}{\lambda} = \left( \frac{2R}{\sum_{w \in W^-} \mu_w} + c' \right)^2 \implies$$

$$\lambda \left( \frac{2R}{\sum_{w \in W^-} \mu_w} + c' \right)^2 - c'^2 \lambda = 4c' \implies$$

$$\lambda = \frac{4c'}{\left( \frac{2R}{\sum_{w \in W^-} \mu_w} + c' \right)^2 - c'^2}$$

Plugging this and $\Delta_w = c'\mu_w$ back into Equations 22, we get for all $w \in W^-$

$$\rho_w = \mu_w \left( \frac{-c' + \sqrt{\dfrac{c'^2 \left( \dfrac{4c'}{\left( \frac{2R}{\sum_{w \in W^-} \mu_w} + c' \right)^2 - c'^2} \right) + 4c'}{\left( \frac{2R}{\sum_{w \in W^-} \mu_w} + c' \right)^2 - c'^2}}}{2} \right)$$

$$= \mu_w \left( \frac{-c' + \sqrt{\dfrac{4c' \left( \left( \dfrac{c'^2}{\left( \frac{2R}{\sum_{w \in W^-} \mu_w} + c' \right)^2 - c'^2} \right) + 1 \right)}{\left( \frac{2R}{\sum_{w \in W^-} \mu_w} + c' \right)^2 - c'^2}}}{2} \right)$$

$$= \mu_w \left( \frac{-c' + \sqrt{\dfrac{\left( \frac{2R}{\sum_{w \in W^-} \mu_w} + c' \right)^2 \left( \left( \frac{2R}{\sum_{w \in W^-} \mu_w} + c' \right)^2 - c'^2 \right)}{\left( \frac{2R}{\sum_{w \in W^-} \mu_w} + c' \right)^2 - c'^2}}}{2} \right)$$

$$= \mu_w \left( -c' + \frac{R}{\sum_{w \in W^-} \mu_w} + c' \right)$$

$$= \frac{\mu_w R}{\sum_{w \in W^-} \mu_w}$$

∎