# OpenReview forum: "Staying up to Date with Online Content Changes Using Reinforcement Learning for Scheduling"
_ICML.cc/2019/Workshop/RL4RealLife — RL4RealLife 2019_

### Official Review · AnonReviewer1 · 2019-05-24
**Accept. Good problem. Large-scale empirical evaluation. A large portion of the paper is not on the main topic of the workshop.**

**Rating:** 4
**Confidence:** 4

**Review:**

This paper proposes a reinforcement learning algorithm for freshness crawling. Freshness crawling is a process where the agent crawls online sources, such as news feeds, that have likely changed since the last crawl. This problem is challenging because the agent does not know whether the source has changed until it is crawled. An important aspect of this problem is that there is a budget on the number of crawls. The authors study several policies for this problem and then propose a learning variant of it.

This paper studies an important problem and the proposed approach is comprehensively evaluated on 20+ million URLs that were crawled for 14 weeks. My detailed comments are below:

1) A large portion of the paper, 3.5 pages out of 8, is devoted to studying properties of the proposed policies. There is really no learning aspect there, which is the topic of this workshop. This part of the paper is not easily readable.

2) I am not convinced that the problem in this paper is a reinforcement learning problem. Why not a structured bandit? Think of this as follows. Suppose that there are two equally important news feeds, feed 1 that is updated once per day and feed 2 that is updated once per week. Suppose that the crawling agent has a budget of 8 crawls per week. Then, on average, the best strategy seems to be to crawl feed 1 once per week and feed 2 once per day. Note that there is no planning aspect here.  When the update rates are unknown, you can formulate the problem of learning the vector of rates (arm) as a bandit problem. Based on your model, where you get individual observations for each crawl, I would argue that either a linear bandit or a combinatorial semi-bandit should work for you,

  https://papers.nips.cc/paper/4417-improved-algorithms-for-linear-stochastic-bandits

  http://proceedings.mlr.press/v38/kveton15.html

---

### Official Review · AnonReviewer2 · 2019-05-25
**An interesting real-world problem**

**Rating:** 4
**Confidence:** 3

**Review:**


The paper considers a very interesting real-world application, i.e., learning the policy for scheduling freshness crawl under bandwidth constraints.

Besides the algorithm under planning setting, the paper proposed model-based reinforcement learning for this problem and test  on a set of 20M  web pages for 14 weeks. The proposed algorithm achieves better performance.

The only concern I have is after the assumptions about the model, the problem eventually becomes bandit problem because of the memoryless property. Why not treat it as bandit problem with constraints?

Considering the bandit problem is a special case of MDP, the paper is still fit to the workshop topic: RL for real-life.

---

### Decision · Program_Chairs · 2019-05-28

Accept